# Modulating inherent lewis acidity at the intergrowth interface of mortise-tenon zeolite catalyst

Huiqiu Wang [1,3], Boyuan Shen[1,2,3], Xiao Chen [1✉], Hao Xiong [1], Hongmei Wang[1], Wenlong Song[1], Chaojie Cui[1], Fei Wei [1✉] & Weizhong Qian [1✉]

The acid sites of zeolite are important local structures to control the products in the chemical conversion. However, it remains a great challenge to precisely design the structures of acid sites, since there are still lack the controllable methods to generate and identify them with a high resolution. Here, we use the lattice mismatch of the intergrown zeolite to enrich the inherent Lewis acid sites (LASs) at the interface of a mortise-tenon ZSM-5 catalyst (ZSM-5-MT) with a 90° intergrowth structure. ZSM-5-MT is formed by two perpendicular blocks that are atomically resolved by integrated differential phase contrast scanning transmission electron microscopy (iDPC-STEM). It can be revealed by various methods that novel framework-associated Al ($Al_{FR}$) LASs are generated in ZSM-5-MT. Combining the iDPC-STEM results with other characterizations, we demonstrate that the partial missing of O atoms at interfaces results in the formation of inherent $Al_{FR}$ LASs in ZSM-5-MT. As a result, the ZSM-5-MT catalyst shows a higher selectivity of propylene and butene than the single-crystal ZSM-5 in the steady conversion of methanol. These results provide an efficient strategy to design the Lewis acidity in zeolite catalysts for tailored functions via interface engineering.

[1] Beijing Key Laboratory of Green Chemical Reaction Engineering and Technology, Department of Chemical Engineering, Tsinghua University, 100084 Beijing, China. [2] Institute of Functional Nano & Soft Materials (FUNSOM), Jiangsu Key Laboratory for Carbon-Based Functional Materials & Devices, Soochow University, 199 Ren'ai Road, 215123 Suzhou, Jiangsu, PR China. [3] These authors contributed equally: Huiqiu Wang, Boyuan Shen. ✉email: chenx123@tsinghua.edu.cn; wf-dce@tsinghua.edu.cn; qianwz@tsinghua.edu.cn

Zeolite is a class of typical crystalline microporous materials consisting of $TO_4$ ($T$ = Si, Al, P) tetrahedra framework. Aluminosilicate zeolites can be used as important solid acid catalysts in a wide range of catalytic applications due to active Brønsted acid sites (BASs) and Lewis acid sites (LASs)[1,2]. Methanol to hydrocarbons (MTH) is one of the most important applications of zeolite-type catalysts. Based on the hydrocarbon pool (HP) mechanism, different products can be obtained from an olefins-based circle and an aromatics-based cycle running in the HP[3–5], affected by the tunable acidic sites (density, type, and distribution) and pore structures. Besides the common BASs, LASs also strongly affect the production of light olefins and aromatics. For example, Lewis acidic $[M(\mu\text{-OH})_2M]^{2+}$ ($M$ = Ca, Mg, and Sr) species formed by incorporating alkaline-earth metals in zeolite catalysts will increase the reaction barriers of benzene methylation and destabilize typical cyclic carbocations in the aromatics-based cycle for a higher propylene selectivity[6,7]. Modulating Lewis acidity in zeolites will efficiently adjust the contribution of two cycles to obtain target products, since catalytically active acid sites in zeolite catalysts play an important role in determining the local concentrations and activities of hydrocarbon species[3,6–10]. Al LASs can be constructed in zeolites with near all kinds of zeolite topologies, which are classified into conventional extra-framework Al ($Al_{EF}$) and inherent framework-associated Al ($Al_{FR}$) LASs based on their distinct structures[11,12]. The $Al_{EF}$ LASs can be generated by removing Al atoms from zeolite frameworks in post-treatments, including steaming and acid or base leaching[13–16]. However, the controllable synthesis of the $Al_{FR}$ LASs has not been achieved to date. Here, we propose that the mismatch at the interface of the intergrown zeolite will cause the missing of O atoms and generate the inherent $Al_{FR}$ LASs. That is, interface engineering still works in porous materials to tailor catalytic performances by designing additional Lewis acidity.

ZSM-5 is a MFI-type zeolite with cross-linked straight and sinusoidal channels, which has been well studied in the MTH catalysis[17–21]. ZSM-5 crystals, as contacted, can form a 90° intergrown structure by connecting straight and sinusoidal channels[22–26]. Such intergrowths will generate a large number of Al LASs at the junctions of different types of channels. However, there is still a lack of the atomic information on zeolite interface, owing to the limits of the low-dose imaging with (scanning) transmission electron microscope ((S)TEM) by their sensitivity to electron beams, the low contrasts of light elements[27–30], and the low availability of atomic-ordered crystals. Recently, the progress of integrated differential phase contrast (iDPC) STEM technique allowed us to achieve the low-dose imaging of various light-element beam-sensitive materials with an ultra-high resolution, such as zeolites and metal-organic frameworks[31,32]. Thus, it is expected that the atomic structures of intergrowth interfaces in ZSM-5 frameworks can be resolved by the iDPC-STEM, which will bring us new understandings of the Lewis acidity at these zeolite interfaces.

In this work, we generate an $Al_{FR}$-LAS-enriched interface in a mortise-tenon ZSM-5 catalyst (ZSM-5-MT) with a 90° intergrown structure. Using the iDPC-STEM, we can reveal the interface structure in the intergrown ZSM-5, and we obviously find the missing of O atoms at this interface which may contribute to the formation of $Al_{FR}$ LASs. Then, we study the enrichment of $Al_{FR}$ LASs in the ZSM-5-MT by different characterization methods, including the $^{27}Al$ solid state nuclear magnetic resonance ($^{27}Al$-NMR) and the Fourier transform infrared (FTIR) spectroscopy of adsorbed probe molecules (pyridine and CO). The catalytic evaluation shows that the ZSM-5-MT can significantly improve the selectivity of propylene and butene in the steady conversion of methanol compared with the conventional ZSM-5 catalysts without intergrown structures. These results bridge the gap between the local structures and catalytic performances of the $Al_{FR}$ LASs at intergrowth interfaces, and provide an efficient strategy for the further design of Lewis acidity.

## Results and discussion

**Morphology of ZSM-5-MT catalyst**. ZSM-5-MT catalyst was synthesized via a one-pot hydrothermal process (as shown in "Methods"). The three-dimensional (3D) morphology of ZSM-5-MT catalyst is revealed by 3D electron tomographic reconstruction in Fig. 1a and Supplementary Video 1. In such ZSM-5-MT, a tenon-like protrusion vertically grows on the (010) surface of an underlying ZSM-5 crystal as a mortise subunit (as shown in the schematic model in Fig. 1b). The morphology of ZSM-5-MT can also be confirmed by annular dark field (ADF) STEM (Fig. 1c-e) and scanning electron microscopy (SEM) (Supplementary Fig. 1, 2) from various projections. Among the ZSM-5-MT samples, around half of crystals have a mortise-tenon architecture (Supplementary Fig. 3). In the X-ray diffraction (XRD) results in Supplementary Fig. 4, we only detect the pure MFI zeolite phase in both ZSM-5-MT and conventional short-b-axis ZSM-5 (ZSM-5-Sb)[33–35]. Thus, the mortise-tenon morphology should be caused only by the intergrowth of ZSM-5 lattices with different orientations[22,36–40]. Consistent pore structures of ZSM-5-MT and ZSM-5-Sb are confirmed by the gas physisorption experiments. As shown in Supplementary Fig. 5 and table 1, there is no evidence that additional mesopores are introduced in the ZSM-5-MT, indicating that the tenon and mortise subunits are closely connected without any gaps.

Then, we used the iDPC-STEM to identify local structures and study such intergrowth behavior in detail. In the iDPC-STEM image obtained from the purple frame in Fig. 1d, we can observe three different areas in Fig. 1f, including pure tenon area, pure mortise area, and intergrowth area. In this projection, the pure tenon area is the typical (010) surface showing the ordered 10-membered rings of straight channels, while the pure mortise area is the (100) surface showing the sinusoidal channels. It indicates that the crystallographic axes **a** and **b** are rotated by 90° around common **c** in space to form the tenon and mortise, respectively. Meanwhile, in the intergrowth area in Fig. 1f, we find the overlapping contrast of (010) and (100) lattice planes where the straight channels are blurred by the perpendicular lattices above. Although the two subunits can not be distinguished in the electron diffraction patterns due to the similar lattice constants of **a** and **b** for ZSM-5 (Supplementary Fig. 6), the sharp spots without splitting or broadening suggest that mortise and tenon are perfectly registered, which is further confirmed by strain analysis across the interface (Supplementary Fig. 7). It means that the mortise and tenon subunits grow into each other to form a perpendicular intergrowth structure just like the structure of a traditional mortise-tenon junction.

We also investigated the connection of channels in tenon and mortise using the images with different defocuses[41]. Figure 1g and h are obtained from the same area marked by the red frame in Fig. 1e but with different defocuses. In Fig. 1g, the (100) surface of mortise subunit is in focus and the sinusoidal channels are clearly imaged in accordance with the structural model, while the (010) surface of tenon subunit with straight channels is in focus in Fig. 1h. Comparing the imaged structures in Fig. 1g, h, it can be deduced that the straight channels in mortise and the sinusoidal channels in tenon are perfectly connected without any dislocations and regardless of local lattice mismatching. We can also confirm such channel connectivity from another projection where the (010) surface of mortise subunit and the (100) surface of

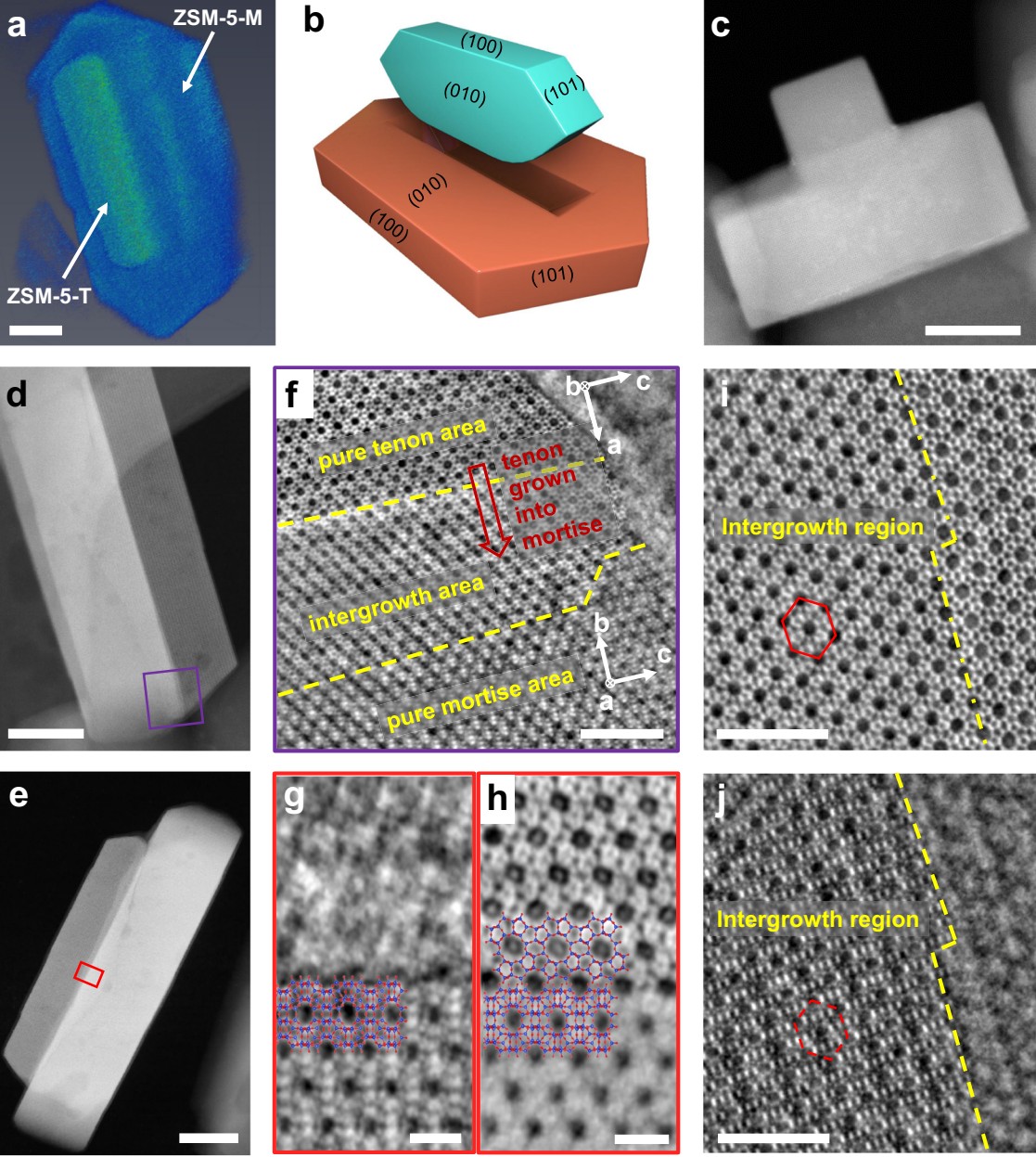

**Fig. 1 Revealing morphology of ZSM-5-MT catalyst. a** 3D electron tomographic reconstruction of individual ZSM-5-MT nanocrystal. **b** Schematic model of ZSM-5-MT nanocrystal with tenon and mortise subunit. **c** ADF-STEM image of ZSM-5-MT from the [001] projection. **d–h** ADF- and iDPC-STEM images revealing the intergrowth structures in ZSM-5-MT from the lateral projection. It is shown that the mortise and tenon grow into each other with perfect connection of different types of channels. The orientation of the crystallographic *a*, *b*, and *c* axes are labeled for two subunits in (**f**). **i, j** Imaging the straight channels in mortise and the sinusoidal channels in tenon at the same area by the iDPC-STEM imaging with different defocuses. Scale bars, 50 nm in (**a**) and (**c-e**), 5 nm in (**f, i**, and **j**), 2 nm in (**g, h**).

tenon subunit can be imaged with different defocuses. As marked by the red hexagons in Fig. 1i, j, the projected positions of the channels in mortise and tenon are coherently located. Moreover, we also revealed other local structures in the ZSM-5-MT, including its surface terminations, surface steps and step-edge sites (Supplementary Fig. 8). These results provide an overall understanding of the morphology of ZSM-5-MT and reveal such intergrowth behavior with confirmed channel connectivity preliminarily.

**Resolving the interface structures of ZSM-5-MT catalyst.** The atomic structures of intergrowth interfaces can be investigated by imaging from the common [001] direction of ZSM-5-MT. From

this projection (Fig. 2a), it is possible to observe intergrowth interfaces to reveal how two areas with different lattice orientations (tenon and mortise) grow into each other. Figure 2b is the iDPC-STEM image obtained from the blue frame in Fig. 2a, which clearly shows the lattice characteristics in this projection. Figure 2c gives the structural models of ZSM-5 with a 90° rotation in its [001] projection, where the straight and sinusoidal channels are marked out, respectively. The green arrows in Fig. 2c show the characteristic pattern of Si-O islands to identify lattice orientations, which can also be observed by the iDPC-STEM and simulation (Fig. 2d). Then, at the intergrowth area, these rotating lattices with two orientations will overlap. If the characteristic arrows in upper and lower lattices are the same, the characteristic

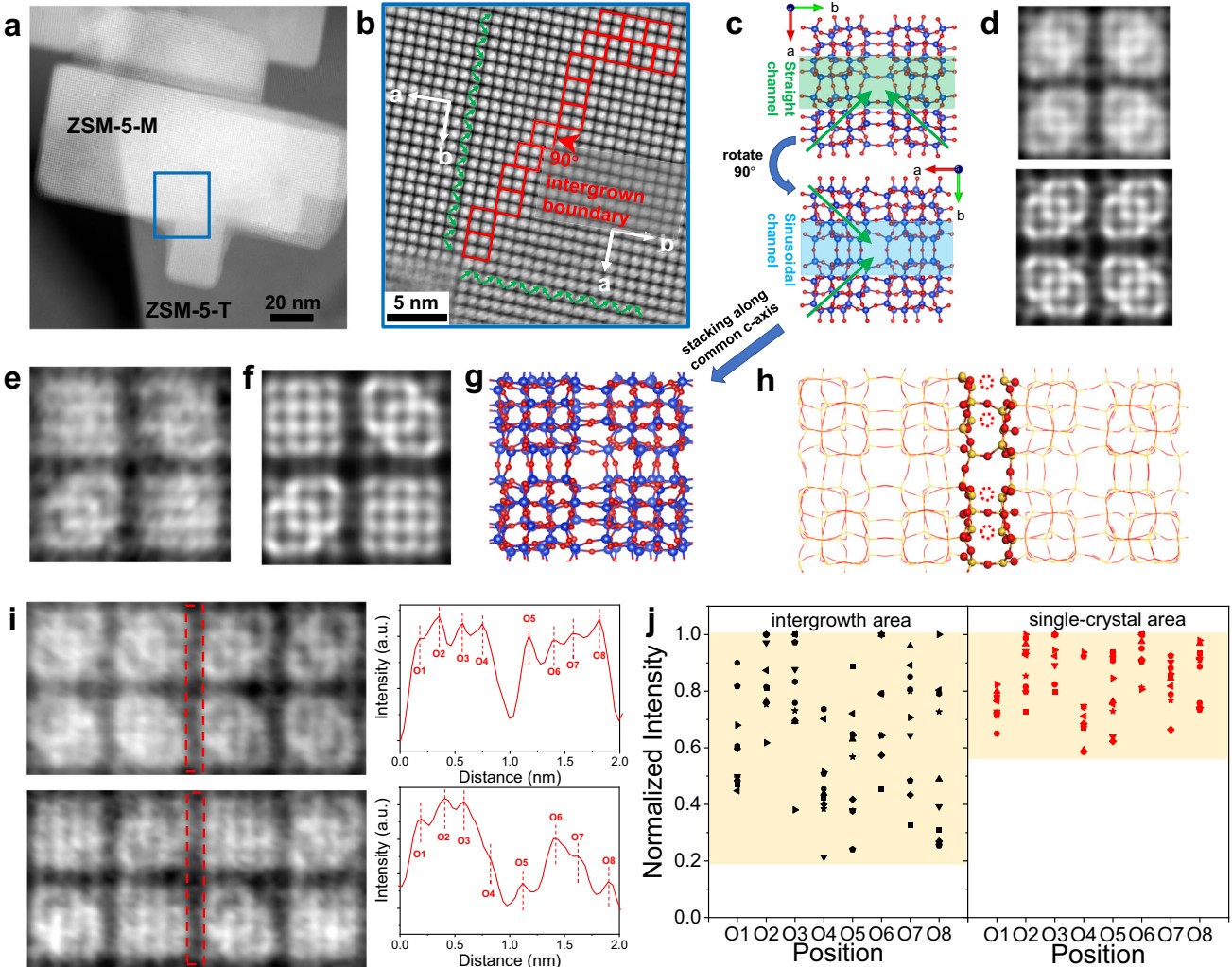

**Fig. 2 Atomic interface structures in ZSM-5-MT catalyst. a** ADF-STEM image of ZSM-5-MT from the [001] projection. **b** Magnified iDPC-STEM image of intergrowth interface in the area marked by the blue frame in (**a**). The intergrowth area between mortise and tenon is marked by red frames. **c** Structural models of ZSM-5 from the [001] projection. **d** Magnified iDPC-STEM image (top) and simulated iDPC-STEM image of ZSM-5 lattice in a pure mortise or tenon area (bottom) from the [001] projection. **e**–**g** Magnified iDPC-STEM image, simulated iDPC-STEM image and structural model of overlapped lattices in the intergrowth area by stacking two models in (**c**) along the *c*-axis of ZSM-5. **h** Structural model of the interface in ZSM-5-MT showing the partial missing of O atoms due to the slight lattice mismatch. **i** Magnified iDPC-STEM images of single-crystal (top) and intergrowth (bottom) areas from the [001] projection and corresponding intensity profiles. Eight positions and intensities of eight O atom columns can be identified in each profile. **j** Statistical results of the normalized intensities of O peaks in single-crystal (red symbol) and intergrowth (black symbol) areas. In statistics, 10 data are included for each area and different shapes of symbols are corresponding to the data in different profiles (Supplementary Figs. 12–14).

patterns are maintained in the overlapped image. If not, a square pattern appears in the overlapped image instead. The imaging results are consistent with the simulation and model in Fig. 2e–g. Based on our analysis of these image characteristics, two areas with different lattice orientations (tenon and mortise) can be identified according to the green arrows in Fig. 2b. The intergrowth interfaces can be outlined by the red frames, where image is just the overlap of the lattice images in two subunits growing into each other.

In this projection, we observed not only the characteristic patterns but also the contrasts of O atoms between them. Based on the structural model in Fig. 2h, there is an inherent missing of O atoms at the interfaces between the 90°-rotation lattices. In order to study such atom missing, we used the profile analysis of the iDPC-STEM images to give the positions and intensities of O peaks as shown in Fig. 2i. The normalized intensities of O peaks can semi-quantitatively reflect the number of atoms in these O atom columns. Based on the image simulation in Supplementary

Fig. 10, the missing of O atoms in columns is accompanied by the decrease of peak intensity quantitatively. We made statistics on the intensities of these O atom columns in single-crystal (pure tenon or pure mortise) and intergrowth areas (more profile data in Supplementary Figs. 12–14), respectively. As shown in Fig. 2j, the O peak intensities are all higher than 0.6 in the single-crystal area, while those in the intergrowth area are varying in a wide range (from 0.2 to 1). It indicates the obvious missing of O atoms at the intergrowth interfaces. It can be interpreted that the increasing distances between Si or Al atoms due to the local lattice mismatch at the interface make them impossible to be bonded by O atoms. These results reveal the interface structures of the tenon-mortise intergrowth, and the non-tetra-coordinated Al atoms formed by the missing of O atoms will generate more Lewis acidity to change the performances of ZSM-5-MT catalyst.

**Acid properties of ZSM-5-MT catalyst**. In order to establish the structure–property relationship of the ZSM-5-MT catalyst, we

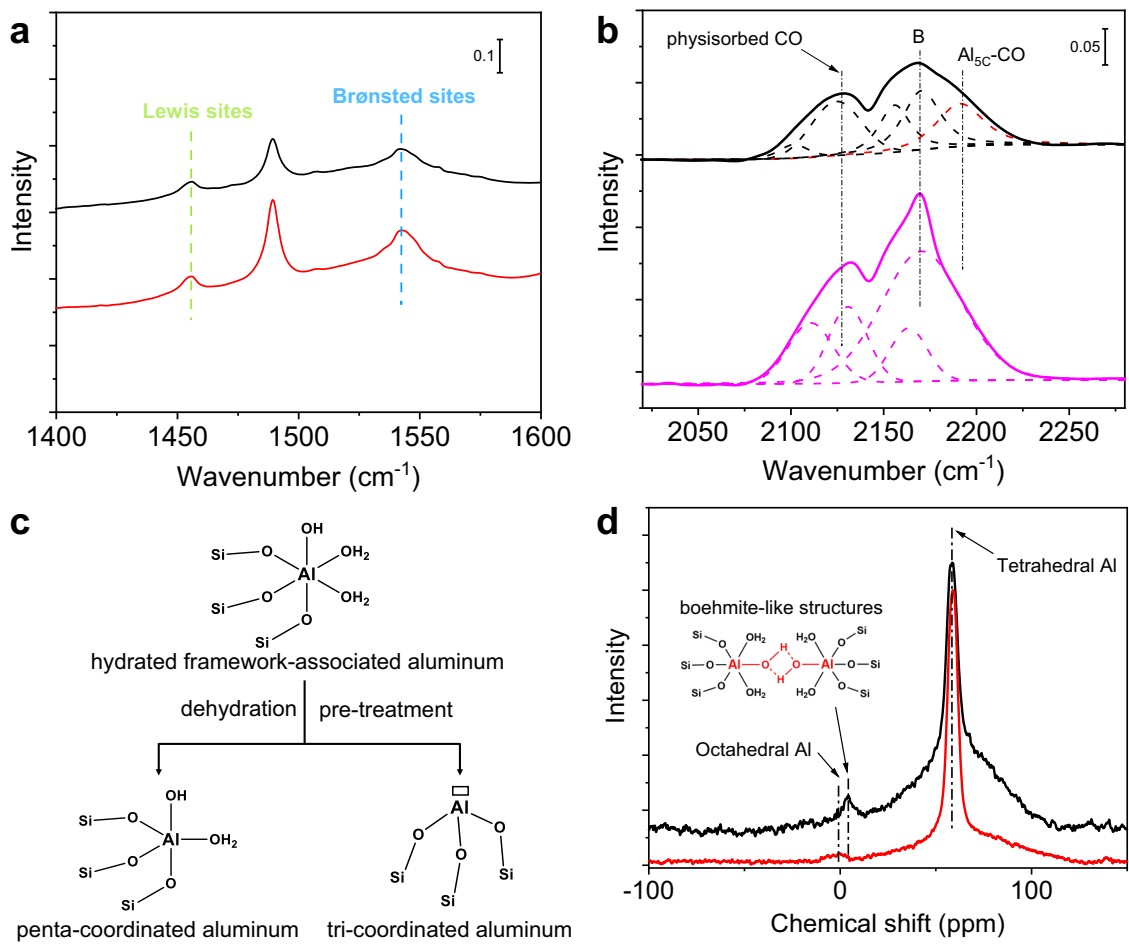

**Fig. 3 Acidity and aluminum characterization of zeolite catalysts. a** FTIR spectroscopy of adsorbed pyridine in ZSM-5-MT (black line) and ZSM-5-Sb-67 (red line). **b** FTIR spectroscopy of adsorbed CO in ZSM-5-MT (black line) and ZSM-5-Sb-67 (pink line). **c** Structural evolution of tri-coordinated $Al_{FR}$ LASs under different conditions. **d** $^{27}$Al MAS NMR of ZSM-5-MT (black line) and ZSM-5-Sb-67 (red line).

studied the acid properties of ZSM-5-MT compared with conventional ZSM-5-Sb without intergrown structure. First, the acid property was investigated by the temperature-programmed $NH_3$ desorption ($NH_3$-TPD) in Supplementary Fig. 16. The $NH_3$-TPD results show that the total acid sites density decreased over ZSM-5-MT compared to ZSM-5-Sb-67, implying that some acid sites were changed by the introduction of the mortise-tenon interface. Then, the acid sites of ZSM-5 zeolites are detected by Fourier transform infrared (FTIR) spectroscopy of adsorbed pyridines and carbon monoxides (CO). These two different probe molecules were used together for deeper knowledge of the nature of acid sites. The FTIR spectra using pyridines as probe molecules are shown in Fig. 3a. The bands at 1455 cm$^{-1}$ and 1545 cm$^{-1}$ can be ascribed to coordinately bond pyridines (pyridines interacting with LASs) and pyridinium ions (pyridines interacting with BASs), respectively[15,42,43]. As we summarized in Supplementary Table 2, the concentration of BAS of ZSM-5-Sb-67 (207.8 μmol/g) is higher than that of ZSM-5-MT (167.6 μmol/g), and the concentration of LAS of ZSM-5-Sb-67 (17.6 μmol/g) is slightly higher than that that in ZSM-5-MT (14.2 μmol/g), while their ratio of BASs/LASs is equal. Although the concentrations of LAS of ZSM-5-MT and ZSM-5-Sb-67 zeolite are nearly equal, their Lewis acidic Al species may be different.

To finely distinguish between multiple LASs, we further detected these acid sites using CO as probe molecules for the FTIR spectroscopy (Fig. 3b). The peaks at 2100–2129 cm$^{-1}$ indicate the physisorbed CO and the peaks at 2150–2170 cm$^{-1}$

results from the CO interacting with the BASs in zeolites[44]. Interestingly, the ZSM-5-MT shows a band at ~ 2193 cm$^{-1}$, which is attributed to the CO interactions with penta-coordinated after the dehydration of zeolites during the pretreatment of the sample (Fig. 3c)[44], while the ZSM-5-Sb-67 does not. Such Al-CO interactions can be promoted by the partial de-coordination of water from the fully hydrated tri-coordinated aluminum species. Therefore, appreciable amounts of the tri-coordinated aluminum sites can only be found in ZSM-5-MT zeolites. Combining with the results of adsorbed pyridine in FTIR spectra, it can be concluded that some tetra-coordinated aluminum species transformed into tri-coordinated aluminum species due to the inherent missing of O atoms at the tenon-mortise interfaces, which agrees with the decrease of the concentration of BAS.

To verify Al sites in terms of their coordination, $^{27}$Al magical-angle-spinning (MAS) nuclear magnetic resonance (NMR) was performed under ambient conditions without pre-treatment (dehydration). The $^{27}$Al MAS NMR spectra (Fig. 3d) of both ZSM-5-MT and ZSM-5-Sb show sharp peaks at 58 ppm, associated with the presence of tetra-coordinated Al in the bulk framework. It validates that most of Al atoms are incorporated into the framework. Meanwhile, the ZSM-5-Sb shows a broad peak of chemical shift at 0 ppm belongs to distorted octahedral Al species, which is experimentally assigned to extra-framework Lewis acidic Al species[7]. As for the ZSM-5-MT, the broad peak at 0 ppm disappeared, while a new peak at 4 ppm corresponding to the boehmite-like structures (given in Supplementary Fig. 18)[45]

arose. In other words, there is almost no detectable extra-framework aluminum species in ZSM-5-MT. Meanwhile, such boehmite-like structures are formed only when the high-density fully hydrated tri-coordinated LASs are very close in space to connect, for example, when such $Al_{FR}$ LASs are concentrated at the interface of zeolite intergrowth. Based on the characterization results of FTIR and $^{27}Al$ MAS NMR, the inherent tri-coordinated $Al_{FR}$ LASs caused by the missing of O atoms in ZSM-5-MT are confirmed experimentally, which is also consistent with the imaging results above. The mortise-tenon interface in ZSM-5-MT results in the formation of $Al_{FR}$ Lewis acid sites as well as the decrease of the concentration of BAS. Overall, we, for the first time, proposed an efficient method to design the novel framework-associated Al LASs of zeolites via interface engineering.

**Catalytic performances of ZSM-5-MT catalyst.** To address the effect of these $Al_{FR}$ Lewis acid sites, we tested the catalytic performances of ZSM-5-MT and ZSM-5-Sb-75 catalysts with the same concentration and strength of acid sites in the conversion of methanol. The selectivities of gas products in the initial and the sufficient stage of reaction are given in Fig. 4a–d, which indicates that two catalysts with different architectures exhibit quite different catalytic performances.

In detail, the initial stage of the reaction is determined as the conversion of methanol is lower than 70–80%, considering the easy activation and easy reaction of methanol with zeolite. After

that, it belongs to the sufficient stage of the reaction. More light alkanes ($C_{1-4}$ alkanes) were formed over ZSM-5-MT compared with ZSM-5-Sb-75 (Fig. 4a), while aromatics in nearly the same amount were formed over two catalysts (Fig. 4b) when the conversion of methanol is lower than 75%. During this stage, a majority of hydrogen transfer products are formed by methanol-induced hydrogen transfer (MIHT)[46]. The generation of aromatics is related to both BASs and LASs, while the light alkanes are formed over LASs[46]. Considering their nearly identical acid sites concentrations, we concluded that the $Al_{FR}$ LASs exhibit higher capacity for MIHT compared to conventional $Al_{EF}$ LASs.

At the sufficient stage of reaction, the selectivities of propylene (44.3%) and butene (16.3%) over ZSM-5-MT are higher than those over ZSM-5-Sb-75 (35.8 % and ~8%, respectively) (Fig. 4c-e). Meanwhile, the ZSM-5-MT shows lower selectivities of ethene (10.2%) and aromatics (11.5 %) than the ZSM-5-Sb-67 (16.6 % and ~19.6 %, respectively). As mentioned above, the olefins-based and the aromatics-based cycles existed in the steady conversion of methanol inside zeolite channels, following the HP mechanism. Almost equal amounts of ethene and propylene were produced in the aromatics-based cycle, while the olefins-based cycle favored the production of much higher yield of propylene than ethene[47–49]. Apparently, the intergrown structure of ZSM-5-MT exhibits the space confinement effect on inhibiting the aromatics-based cycle, while amplifying the olefins-based cycle. In addition, the higher ratio of propylene to propane and higher ratio of butene to butane were retained with ZSM-5-MT as compared to

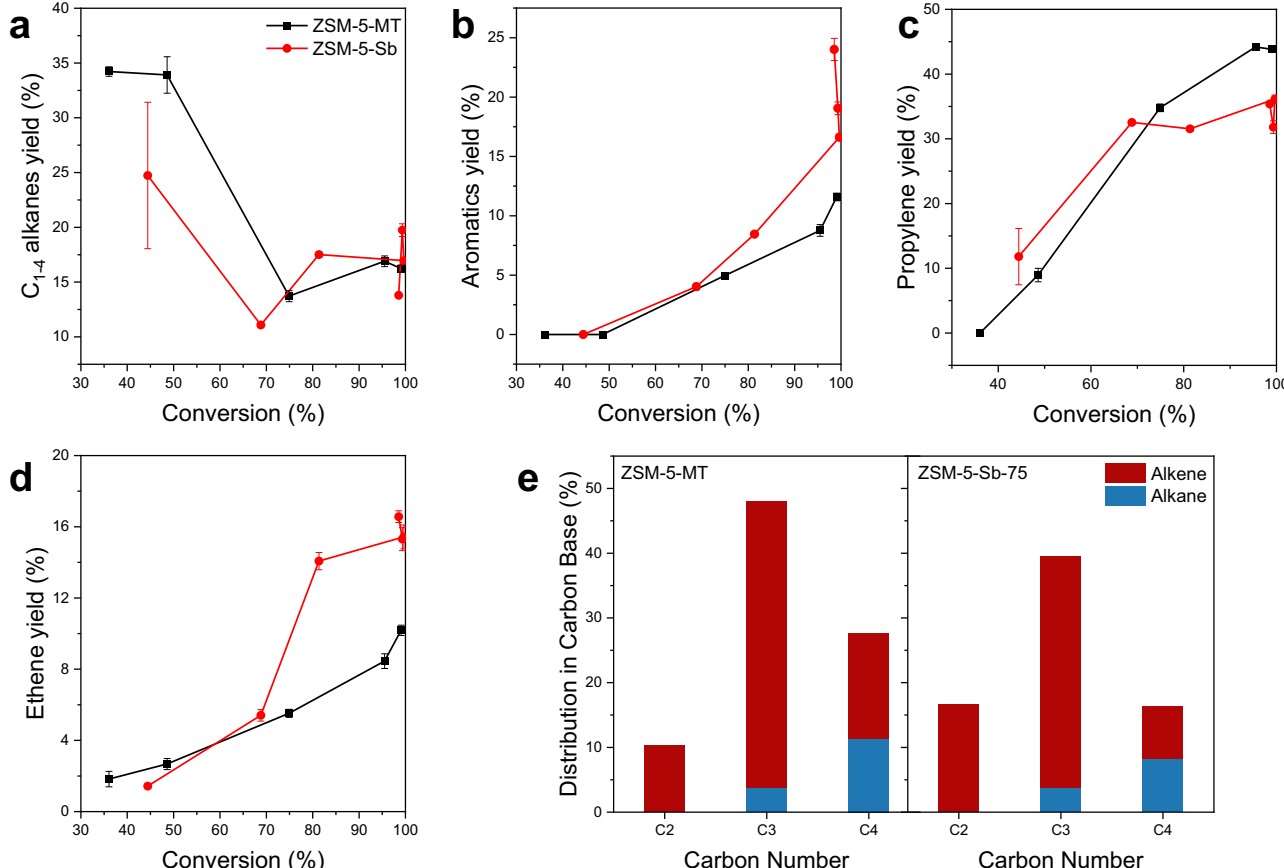

**Fig. 4 Catalytic performance of ZSM-5-MT catalyst.** Yields of light alkanes (**a**), aromatics (**b**), propylene (**c**), and ethene (**d**) as a function of the conversion of methanol with ZSM-5-MT and ZSM-5-Sb-75 catalysts. **e** Distribution of different carbon numbers of hydrocarbon products for ZSM-5-MT and ZSM-5-Sb-75 catalysts at complete methanol conversion (≥99%) (Test conditions: 475 °C, weight hourly space velocity (WHSV) of 3 h$^{-1}$). The error bars in **a** to **d** represent the standard deviations of three sets of data in repeated experiments.

ZSM-5-Sb. These further confirm that the $Al_{FR}$ LASs of ZSM-5-MT suppress the hydrogen transfer effect significantly. Noted here that the present sample contained only half mortise-tenon architectures (see Supplementary Fig. 3), it can be expected that pure mortise-tenon architecture will be favorable to produce propylene and butene with much higher selectivity.

In summary, we successfully modulated the Lewis acidity of zeolite catalyst by creating a mismatched interface in the intergrown structures. In a mortise-tenon ZSM-5 intergrowth, the lattice mismatch on the interface will generate the missing of O atoms to form inherent $Al_{FR}$ LASs. Using the iDPC-STEM, we can atomically resolve these local structures in real space to confirm the missing of O atoms on the interface. Then, we characterized these abundant $Al_{FR}$ LASs in ZSM-5-MT by different macroscopic methods, including $^{27}Al$ MAS NMR and FTIR spectroscopy of adsorbed probe molecules. Combining these methods together, we find that the tri-coordinated Al sites caused by the missing of O atoms contribute to the generation of $Al_{FR}$ LASs in ZSM-5-MT. Based on the catalytic test, the resulting $Al_{FR}$ LASs in ZSM-5-MT catalysts significantly promote the propagation of the olefins-based cycle, exhibiting the higher selectivities of propylene and butene in steady MTH, and showing a higher capacity for MIHT compared to conventional $Al_{EF}$ LASs. These results provide a general method of interface engineering to design the zeolite structures for the enhanced Lewis acidity, and provide new insights into the structure–property relationship of zeolite catalysts from the perspectives on atomic local structures.

## Methods

**Synthesis of ZSM-5-MT and ZSM-5-Sb crystals**. ZSM-5-MT and ZSM-5-Sb were both synthesized by a conventional hydrothermal method. Tetraethyl orthosilicate (TEOS), $Al(NO_3)_3·9H_2O$ and tetrapropylammonium hydroxide (TPAOH) were used as the silicon source, aluminum source, and structure-directing agent, respectively. In a typical run of the preparation of ZSM-5-Sb samples, 26.2 g TPAOH (wt 25%), 22.4 g TEOS, 4.0 g urea, 0.6 g $Al(NO_3)_3·9H_2O$, 0.2 g NaOH and 0.2 g iso-propanol (IPA) were added into 36.8 g $H_2O$ under stirring. After being stirred for 2 h at room temperature, the gel was transferred into a 200 mL stainless steel autoclave with a Teflon liner. The gel was heated from 30 °C to 180 °C at a rate of 6.25 °C/h, and then held for 2 days at 180 °C. Subsequently, the autoclave was quenched with cold water. The solids were filtered, washed three times with deionized water, dried at 110 °C in air, then calcined at 550 °C for 6 h to remove the organic templates.

For the synthesis of ZSM-5-MT samples, 26.2 g TPAOH (wt 25%), 22.4 g TEOS, 4.0 g urea, 0.6 g $Al(NO_3)_3·9H_2O$, 0.6 g NaOH and 0.2 g iso-propanol (IPA) were added into 36.8 g $H_2O$ under stirring. After being stirred for 2 h at room temperature, the gel was transferred into a 200 mL stainless steel autoclave with a Teflon liner. The gel was heated from 30 °C to 180 °C at a rate of 3.125 °C/h, and then held for 2 days at 180 °C. The product was extracted from mother liquor using the procedure described above.

In addition, all the Na-type samples were converted into the H-type samples by three-time ion exchanges with 1 M $NH_4NO_3$ solution and subsequent calcination in air at 550 °C for 5 h.

**Imaging conditions and electron tomography experiments**. The high-resolution iDPC-STEM and ADF-STEM images were obtained under a Cs-corrected STEM (FEI Titan Cubed Themis G2 300) operated at 300 kV. The STEM was equipped with a DCOR + spherical aberration corrector for the electron probe which was aligned using a standard gold sample before observations. The aberration coefficients we used were shown as following: $A1 = 0.982$ nm; $A2 = 5.79$ nm; $B2 = 20.4$ nm; $C3 = −250$ nm; $A3 = 137$ nm; $S3 = 89.3$ nm; $A4 = 3.4$ μm, $D4 = 2.25$ μm, $B4 = 1.81$ μm, $C5 = 77.4$ μm, $A5 = 175$ μm, $S5 = 118$ μm, and $R5 = 5.78$ μm. The convergence semi-angle was 15 mrad. And four images used for 2D integration were acquired by a 4-quadrant DF4 detector with an optional high-pass filter applied to reduce the low frequency information in the image. The beam current was reduced lower than 0.1 pA. The collection angles of iDPC-STEM imaging were set as 4–22 mrad. The electron tomography experiments were performed using a Fischione 2020 advanced tomography holder aided with Tomography STEM software (Thermo Fisher Scientific). Tilt images were obtained at the range of ±60° with interval of 2° at low tilt (−52° to 50°) and 1° at high tilt. The commercial Inspect3D software (Thermo Fisher Scientific) was used for 3D tomographic reconstruction. The experimental iDPC-STEM images were filtered by low-pass gaussian blur for noise smoothing.

The iDPC-STEM image simulations were carried out on the basis of the multi-slice approach[50], which can be extended to support iDPC-STEM as explained in the previous works[51–53]. The parameters for simulations, such as convergence semi-angle (15 mrad), collection angle (4–22 mrad), aberration coefficients and applied dose, were adopted from our imaging experiments. During the simulations, four-sector ADF images were obtained from four segments of the four-quadrant detector spanning 4–22 mrad and then applied to generate the iDPC images using the approach described in the literature[51]. The final iDPC-STEM images were convolved with a two-dimensional Gaussian function of 80-pm full-width at half maximum to account for finite probe size.

**Other characterizations**. The scanning electron microscope of samples was collected using JEOL JSM-7401. Textural analysis of zeolite samples was performed by Ar adsorption/desorption using a Micromeritics ASAP 2460 gas sorptiometer and $N_2$ adsorption/desorption using an Autosorb-iQ2-C system (Quantachrome Instruments). By using the t-plot method, the micropore surface area of samples was determined from Ar adsorption isotherm. Moreover, micropore volume and micropore size were determined from the adsorption branches of Ar isotherms with relative pressure p/p0 of <0.01. The Brunauer-Emmett-Teller method was used to determine mesopore surface area from $N_2$ adsorption isotherms. X-ray diffractions (XRD) were recorded on a Rigaku D/Max-RB diffractometer with Cu Kα Radiation at 40 kV and 120 mA to verify the crystalline structure. $^{27}Al$ MAS NMR experiments were performed on a JNM-ECZ600R spectrometer at resonance frequencies of 156.4 MHz. One-dimensional single-pulse $^{27}Al$ MAS NMR spectra were collected on a 3.2 mm probe with a spinning rate of 12 kHz. $Al(NO_3)_3$ was used as the reference of chemical shift at 0 ppm. The Si/Al ratio of the samples was analyzed by inductively coupled plasma optical emission spectrometer (ICP-OES, SPECTRO ARCOS). $NH_3$ temperature-programmed desorption (TPD) experiments were completed in a Quantachrome automated chemisorption analyzer.

**FTIR spectroscopy**. The low temperature IR spectra of CO adsorbed over zeolites were recorded using a Thermo Nicolet iS20 FTIR spectrometer equipped with KBr windows, a MCT-A detector and HARRICK in situ reaction chamber. Prior to the measurements, the sample (200 mg) was loaded into the in situ reaction chamber and pretreated in a flow (25 mL/min) of $N_2$ at 450 °C for 1 h. Then, the chamber was cooled to −95 °C for carbon monoxide adsorption measurements. The background spectrum was collected in the continuous $N_2$ flowing. Subsequently, a mixture gas of 1 vol% $CO/N_2$ (25 mL/min) was introduced into the reaction chamber, and the spectra were collected with time until there was no change in the signals.

The IR spectra of pyridine adsorbed over zeolites were recorded using a Thermo Nicolet iS50 FTIR spectrometer equipped with a DTGS detector and KBr windows. About 100 mg of the sample was pressed into a self-supporting wafer and activated in vacuum ($<10^{-4}$ Pa) at 500 °C for 1 h. For pyridine adsorption, the samples were exposed to pyridine for 1 h after cooling to 100 °C, then the IR transmission cell was evacuated to vacuum (about $10^{-4}$ Pa). The spectra were collected at 200 °C. For the quantitative comparison, the concentration of Brønsted and Lewis acid sites was calculated based on the IR band area at 1515–1565 $cm^{-1}$ and 1430–1470 $cm^{-1}$ after evacuation at 200 °C, using the molar integral extinction coefficients of 1.88 cm/μmol Brønsted acid sites and 1.42 cm/μmol for Lewis acid sites.

**Catalytic performance tests**. The methanol-to-hydrocarbon (MTH) reaction was performed using a quartz fixed-bed continuous-flow reactor (inner diameter = 10 mm) equipped with gas chromatography (GC-2014, equipped with two flame ionization detectors (FIDs) and a thermal conductivity detector (TCD), Shimadzu Co.), catalyst (0.3 g H-form zeolite, homogeneously diluted with 1 g of quartz sand) and a reaction temperature of 475 °C. $CH_3OH$ (99.9% purity) was continuously pumped into the reactor at a WHSV of 5 h by a plunger pump (Labaliance Series II) at 0.03 mL/min with a preheat inert gas of nitrogen (15 mL/min). The effluent from the reactor was maintained at 200 °C and analyzed online by a gas chromatography. All the data discussed in article were collected after 5 h of MTH reaction. For MTH at partial conversion, catalyst loading was adjusted to achieve a wide range of methanol conversion and contact time, and every data point was measured over a fresh sample.

## Data availability

The authors declare that all relevant data supporting the findings of this study are available within the paper and its Supplementary Information files. Additional data are available from the corresponding authors upon reasonable request.

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

## Acknowledgements

This work was supported by the National Key Research and Development Program of China (No. 2020YFB0606401) and (2018YFB0604801), and the National Natural Science Foundation of China (22005170). B.S. thanks the support from Collaborative Innovation Center of Suzhou Nano Science & Technology, the 111 Project, Joint International Research Laboratory of Carbon-Based Functional Materials and Devices.

## Author contributions

H.W., X.C., and W.Q. conceived this project and designed the studies; C.C., F.W., and W.Q. supervised the project. H.W. and X.C. performed the electron microscopy experiments and data analysis; H.W. prepared and characterized the zeolite samples; H. X. performed the simulation of iDPC-STEM images; H.X., H.-M.W., and W.S. helped on the data analysis. H.W., B.S., and X.C. wrote the paper with the contributions from all authors.

## Competing interests

The authors declare no competing interests.
