## [Peer Review File · Nature Communications]

Title: Modulating Inherent Lewis Acidity at the Intergrowth Interface of Mortise-Tenon Zeolite CatalystREVIEWER COMMENTS

Reviewer #1 (Remarks to the Author):

The paper reports a Mortise-Tenon zeolite structure and claim the control of Lewis acid sites by forming such intergrowth interface. The authors took some nice images using iDPC-STEM that is expected to be sensitive to light elements. However, their proposed structure models and the idea to control the acid sites are not well supported by their results. Specific comments are listed below:

1. To fully validate the MT structural model, the tenon tongue should be proven to physically enter the mortise hole by morphological analysis. In the meanwhile, the two components should be proven to have fixed but different crystal orientations. These points are not well addressed in the paper.
2. It does not make sense to employ electron tomography for validating the MT structure because the two components are completely indistinguishable from each other by intensity. It is not possible to rule out that the two crystals just simply stick to each other. Electron diffraction or rotation electron diffraction (RED) should be a better and easier way to identify the 3D heterostructure and also to screen the crystals.
3. The two components of MT structure would exhibit large difference in height from either top or side views of the heterostructure in Fig. 1. It may cause problems for STEM imaging because it suffers from a very small focal depth. In this case, the MT interface cannot be properly visualized by in-focus images.
4. As iDPC is a typical phase contrast imaging technique, the authors should be careful with the interpretation of iDPC contrast variation with respect to the sample height and thickness. For two neighboring components with very different thickness or height, the iDPC contrast may not be directly compared. For example, Fig. 1i and h shows very different “in-channel contrast” for two different regions featuring different height. The authors should systematically investigate the effects of sample thickness/height and imaging conditions on the iDPC contrast to rule out any possible artefacts. In addition, a full comparison between experimental and simulated images/FFTs is necessary to validate the proposed MT model.
5. The authors try to quantify oxygen vacancies by iDPC at the MT junction in Fig. 2, which seems to be questionable. The quantitative evaluation of oxygen vacancy at least requires that i) the iDPC contrast is quantitative, ii) the iDPC image resolution is sufficient to resolve individual oxygen columns, iii) the intensity at MT interface is not from either distorted framework or image distortion/noise, and iv) the intensity variation is not from the disorder of oxygen atoms at the MT junction. However, these points are not mentioned at all in the paper.
6. TEM used in this paper has poor sampling capability. If the materials are used for catalytic applications, the yield of MT structures should be accurately evaluated.
7. The MT structure is very unique and the growth mechanism should be investigated.
8. The authors have claims such as “...atomic interface structure, line 75”, “...tenon-mortise intergrowth in atomic precision, line 179”, which seem to be overstated. The two components of the heterostructure differ in height and cannot be imaged with an atomically-resolved interface.
9. The discussion about contents of Fig. 1g and h is confusing. From line 128, the lower part of Fig. 1g is in-focus with sinusoidal channels, which seems to contradict with the labels in Fig. 1h.

Reviewer #2 (Remarks to the Author):

The manuscript sets forth a problem that has been already discussed in the literature: the role of Lewis acid sites in the methanol-to-hydrocarbons reaction. Other excellent works help to clarify this topic: <https://doi.org/10.1038/s41557-018-0081-0>, <https://doi.org/10.1021/jacs.9b07484>, <https://doi.org/10.1021/jacs.6b09605>. Basically, modulating the concentration of Lewis acid sites (LAS) and Brønsted acid sites (BAS) changes the BAS density, which determines the catalytic performance. Making the solution more complex than the problem does not greatly help. There are other more efficient and reliable methods to generate LAS and change the acid density of a zeolite. Therefore, I see a lack of novelty in the overall research work and the conclusions are not original (as the authors' observations is a well-known result from other published works). Thus, I think this work is not suitable for publication in Nature Communications, though I consider this work may be interesting for publication in other journal upon making several changes.

The following comments also justify my decision, and I encourage the authors to resolve them in order to improve their work:

1. The acid properties of the zeolites used in this work are key to prove the authors' conclusions. For this, the use of NH₃ and pyridine adsorption is a routine analysis for zeolites and the tests are very standardized. Thus, I see several deficiencies on the results presentation and discussion when using these probe molecules to measure the zeolite acidity. Nowadays, it is easy to quantify the amount of adsorbed NH₃ and therefore the amount of desorbed NH₃ can be reliably quantified. I do not see the total acidity measured by NH₃ adsorption. Besides, it is unacceptable to present NH₃-TPD of two samples using arbitrary units (using arbitrary units could work for one sample). Additionally, the authors should refer to acid strength when analyzing the TPD profiles. Likewise, the FTIR spectra of pyridine-saturated zeolites are not properly presented, additional treatments can improve the result analysis, such including a deconvolution of the bands. At a glance, the BAS concentration in the ZSM-5-Sb zeolite looks to be lower than that in the ZSM-5-MT (inferred from the peak intensity/area). The authors should check this and improve the correlation of acid properties with the catalyst performance.

2. The catalyst tests are pretty simple and do not greatly support the authors' claims. To better observe kinetic differences among different catalysts, it is interesting to evaluate the conversion and product yield or selectivity at different space-times. You can find a good example of this in these publications: <https://doi.org/10.1021/jp4053677>, <https://doi.org/10.1021/jacs.6b09605>. Varying the contact time, varies the conversion and product yield/selectivity, and what is interesting is to compare the product yield/selectivity at different conversions (different from 100%). Thus, a catalyst would be kinetically different from other if the product yield/selectivity is different at the same conversion (conversion different from 100%). For this, it would be interesting to see curves of product yield/selectivity against conversion. These comparisons provide a better understanding of the catalyst performance differences. Obviously, a catalyst with a lower BAS density operating at 100% conversion would yield more light

olefins than a catalyst with a higher BAS density because light olefins are intermediates in the methanol-to-hydrocarbons reaction. Furthermore, the authors should correlate and quantify the catalyst performance with the acid properties.

Reviewer #3 (Remarks to the Author):

This manuscript describes modulating inherent Lewis acidity at the intergrowth interface of zeolite catalyst, which is very important for zeolite catalysts. In particular, authors show high quality TEM images, which is excellent for understanding the intergrowth interface of zeolite catalysts. Therefore, I strongly recommend to publish this work after minor revisions in the following:

1. As shown in Fig. 3a, if authors want to compare the peak intensity of IR bands, authors should calibrate the sample amount or the IR bands should be calibrated by Si-O band at near 1800 cm^{-1} . Otherwise, it is difficult to compare the IR band intensity.
2. The XRD patterns in the SI are very poor. Authors should offer high quality XRD patterns of the samples.

Response to review comments

Referee #1

We are very grateful to your professional comments on the structure details of our samples. As made further careful observation, simulation and histogram of sample, we provide sufficient data to validate our structure. The quality of the manuscript is indeed improved significantly with your helpful suggestions, which will not only benefits our team, but also wide readers of the journal.

We replied your comments point by point as follows.

Thanks again.

1. To fully validate the MT structural model, the tenon tongue should be proven to physically enter the mortise hole by morphological analysis. In the meanwhile, the two components should be proven to have fixed but different crystal orientations. These points are not well addressed in the paper.

Reply:

Figure R1. SEM images of some individual ZSM-5-MT crystals (marked by red dash frames).

Thank the referee for the constructive comments. Scanning electron microscope (SEM)

and annular dark field (ADF) STEM images were used to perform morphological analysis of the individual ZSM-5-MT crystals. As shown in Fig. R1, a fin-like protrusion vertically grows on the (010) surface of underlying coffin shape crystal in such ZSM-5-MT crystal. From the lateral view (Fig. R1c), the fin-like protrusion and underlying crystal are closely attached without obvious gaps.

Furthermore, we provide more ADF-STEM images (including the images in Fig. 1, Supplementary Fig. 6) to validate the mortise-tenon structural model from morphology. In Fig. R2a, the projected morphology of individual ZSM-5-MT crystal is consistent with the results from the SEM images. More importantly, from the lateral view (Fig. R2b), a gray curve inside the underlying crystal represent the boundary between fin-like protrusion and underlying crystal, which due to the missing of O atoms at the intergrown interfaces. Thus, we can conclude that the tenon tongue physically enter the mortise hole, based on which we proposed the mortise-tenon structural model.

Figure R2. The ADF-STEM images of some individual ZSM-5-MT crystals. The boundary

between tenon and underlying mortise crystal marked by red arrows in magnified image.

In the original version, the crystal orientations of tenon and mortise subunit were determined with the iDPC-STEM imaging from the lateral projection in Fig. R3 (Fig. 1f in the main manuscript). The characteristic (010) surface with the ordered 10-membered rings of straight channels and (100) surface with the sinusoidal channels are clearly imaged in tenon crystal and mortise crystal, respectively. It indicates that the crystallographic axes **a** and **b** are rotated by 90° around common **c** in space to form the tenon and mortise respectively. Therefore, we can use such iDPC-STEM image in Fig. 1f to determine the crystal orientations of tenon and mortise subunit.

In the revised version, more detailed crystallographic orientation marks were added in Fig. 1f. The words of “The crystallographic orientation of the a, b and c axes is labeled for two subunits in f.” was added in the figure caption of Fig. 1. Thus, it can be confirmed that the two components are fixed but with different crystal orientations, as you mentioned.

Figure R3. iDPC-STEM images revealing the intergrowth structures in ZSM-5-MT from the lateral projection. The orientation of the crystallographic **a**, **b** and **c** axes is labeled for two subunits.

These figures and related discussions have been added in the revised version of main manuscript and supplementary information.

2. It does not make sense to employ electron tomography for validating the MT structure because the two components are completely indistinguishable from each other by intensity. It is not possible to rule out that the two crystals just simply stick to each other. Electron diffraction or rotation electron diffraction (RED) should be a better and easier way to identify the 3D heterostructure and also to screen the crystals.

Reply:

Thank you for your comments. In the original version of the manuscript, we mainly employed electron tomography to reveal the three-dimensional (3D) morphology of ZSM-5-MT crystal. The iDPC-STEM imaging from the [001] direction of ZSM-5-MT crystal is used to validate the mortise-tenon structural model and investigate intergrowth interfaces (Fig. R4). The mortise and tenon subunits have coincident c-axes. And, the two components of MT structure exhibit smaller difference in height from [001] direction, which clearly demonstrated in the morphology of ZSM-5-MT crystal.

Figure R4. **a**, ADF-STEM image of ZSM-5-MT from the [001] projection. **b**, Magnified iDPC-STEM image of intergrowth interface in the area marked by the blue frame in **a**. The intergrowth area between mortise and tenon is marked by red frames.

In addition, considering the new images of Figure.R1 and Figure R2 together, these

evidences are sufficient to validate the structure of ZSM-5-MT.

Thanks again.

3. The two components of MT structure would exhibit large difference in height from either top or side views of the heterostructure in Fig. 1. It may cause problems for STEM imaging because it suffers from a very small focal depth. In this case, the MT interface cannot be properly visualized by in-focus images.

Reply:

We agree with the comments of referee. Fortunately, the iDPC-STEM images with different defocuses (Fig. 1g, h, i, and j) allowed us to investigate the connection of channels in tenon and mortise. As illustrated in the response of question #2, the iDPC-STEM imaging from the [001] direction of ZSM-5-MT crystal clearly show the MT interfaces (see Fig. R5). The two components of MT structure exhibit small difference in height from [001] direction.

Figure R5. **a**, ADF-STEM image of ZSM-5-MT from the [001] projection. **b**, Magnified iDPC-STEM image of intergrowth interface in the area marked by the blue frame in **a**. The intergrowth area between mortise and tenon is marked by red frames. **c**, Structural models of ZSM-5 from the [001] projection. **d**, Magnified iDPC-STEM image (top) and simulated iDPC-STEM image of ZSM-5 lattice in pure mortise or tenon area (bottom) from the [001] projection. **e-g**, Magnified iDPC-STEM image, simulated iDPC-STEM image and structural model of overlapped lattices in the intergrowth area by stacking two models in **c** along c-axis of ZSM-5.

From this projection (Fig. R4a), it is possible to observe intergrowth interfaces to reveal how two areas with different lattice orientations (tenon and mortise) grow into each other. Fig. R5b is the iDPC-STEM image obtained from the blue frame in Fig. R5a, which clearly shows the lattice characteristics in this projection. Fig. R5c gives the structural models of ZSM-5 with a 90° rotation in its [001] projection, where the straight and sinusoidal channels are marked out, respectively. The green arrows in Fig. R5c show the characteristic pattern of Si-O islands to identify lattice orientations, which can also be observed by the iDPC-STEM and simulation (Fig. R5d). Then, at the intergrowth area, these rotating lattices with two orientations will overlap. If the characteristic arrows in upper and lower lattices are the same, the characteristic patterns are maintained in the overlapped image. If not, a square pattern appears in the overlapped image instead. The imaging results are consistent with the simulation and model in Fig. R5e-g. Based on our analysis on these image characteristics, two areas with different lattice orientations (tenon and mortise) can be identified according to the green arrows in Fig. R5b. The intergrowth interfaces can be outlined by the red frames. Overlap of the lattice just occurred when two subunits growing into each other.

4. As iDPC is a typical phase contrast imaging technique, the authors should be careful with the interpretation of iDPC contrast variation with respect to the sample height and thickness. For two neighboring components with very different thickness or height, the iDPC contrast may not be directly compared. For example, Fig. 1i and h shows very different “in-channel contrast” for two different regions featuring different height. The authors should systematically investigate the effects of sample thickness/height and imaging conditions on the iDPC contrast to rule out any possible artefacts. In addition, a full comparison between experimental and simulated images/FFTs is necessary to validate the proposed MT model.

Reply:

Thank the referee for the kind suggestions. As illustrated above, the iDPC-STEM images with different defocuses (Fig. 1g, h, i, and j) is just used to study the crystal

orientations of two subunits and the connection of channels in tenon and mortise, not to compare their contrasts quantitatively. The sample height and thickness will not affect our interpretation on crystal orientations and channel connection in Fig. 1g, h, i, and j. A new reference was added to support this point in the revised manuscript, which is: 36. Bosch EGT, Lazić I. Analysis of depth-sectioning STEM for thick samples and 3D imaging. *Ultramicroscopy* **207**, 112831 (2019).

As using the iDPC-STEM images of the [001] projection of the ZSM-5-MT crystal to investigate the oxygen vacancies and MT interfaces (see Fig. R4), the tenon and mortise parts show little difference in thickness (as shown in Fig. R1). Thus, the effect of sample thickness/height on the analysis of image contrast can be ignored. In addition, the intensities of oxygen column peaks in all profile were normalized, allowing a fair comparison on the normalized intensities to investigate the oxygen vacancies.

We also agree that it is necessary for a full comparison between experimental and simulated images. We performed the simulation of iDPC-STEM images based on the same parameters and integration processes that we applied in the imaging experiments. The experimental and simulated images were fully compared to prove that the intergrowth area results from the overlap of tenon and mortise crystal (Fig. R6). As indicated in Fig. R6, the imaging results are perfectly consistent with the simulated results.

Figure R6. **a**, Structural models of ZSM-5 from the [001] projection. **b**, Magnified iDPC-STEM image (top) and simulated iDPC-STEM image of ZSM-5 lattice in pure mortise or tenon area (bottom) from the [001] projection. **c**, Structural model of overlapped lattices in the intergrowth area. **d**, Magnified iDPC-STEM image (top) and simulated iDPC-STEM image of overlapped lattices in the intergrowth area (bottom) from the [001] projection.

area. **d**, Magnified iDPC-STEM image (top) and simulated iDPC-STEM image of ZSM-5 lattice in intergrowth area (bottom) from the [001] projection.

We have added these simulated results and discussions in the revised main manuscript and supporting information.

5. The authors try to quantify oxygen vacancies by iDPC at the MT junction in Fig. 2, which seems to be questionable. The quantitative evaluation of oxygen vacancy at least requires that i) the iDPC contrast is quantitative, ii) the iDPC image resolution is sufficient to resolve individual oxygen columns, iii) the intensity at MT interface is not from either distorted framework or image distortion/noise, and iv) the intensity variation is not from the disorder of oxygen atoms at the MT junction. However, these points are not mentioned at all in the paper.

Reply:

Thank you for your suggestions. Four points above are replied one by one as follows.

i): We performed iDPC-STEM image simulations to investigate the relation between the iDPC contrast and the amount of oxygen molecules. We give the simulated results of several different samples in Fig. R7a, including the perfect ZSM-5 and the defective ZSM-5 with different amounts of oxygen vacancies. The results of profile analysis (Fig. R7b and c) indicate that the height (intensity) of oxygen column peak is linearly related to the amount of oxygen vacancies. Therefore, the iDPC contrast is quantitative for this oxygen column, which can be used to investigate the oxygen vacancies.

Figure R7. **a**, The simulated iDPC-STEM images of five ZSM-5 unit cells with the different amounts of oxygen vacancies (The oxygen atom columns with vacancies are marked by dash red circles). **b**, The intensity profiles of six ZSM-5 frameworks (six color dash arrows shown in the top six images) with different amount of oxygen vacancies. **c**, The contrast intensity of the oxygen atom columns related to the amount of oxygen vacancies.

ii):

In the corresponding FFT pattern of Fig. R8b, the $(-12,7,0)$ plane shows an information transfer of 1.46 \AA , which allows us to resolve individual oxygen columns, since the distances between adjacent oxygen columns are larger than 1.46 \AA . As expected, sixteen individual oxygen columns can be clearly identified by the red arrows in the magnified image (Fig. R8c).

Figure R8. **a**, The iDPC-STEM image of ZSM-5-MT crystal from the [001] direction. **b**, The corresponding FFT pattern of **a** in a log scale. **c**, The magnified iDPC-STEM image of the intergrown area in **a** (individual oxygen columns are marked by red arrows).

iii) and iv):

To avoid the effects of image distortion and noise, the statistical analyses of single-crystal and intergrowth areas (Fig. 2i and j) were performed in the same image (Fig. R8a). As mentioned above, the individual oxygen columns of the intergrown area can be clearly identified in our iDPC-STEM images, which confirms that the intensity variation is not from the disorder of oxygen atoms or distorted framework.

Therefore, these doubts about the image interpretation can be fully addressed, and the reliability of imaging can be guaranteed. The related figures and discussions have been added in the revised main manuscript and supplementary information.

6. TEM used in this paper has poor sampling capability. If the materials are used for catalytic applications, the yield of MT structures should be accurately evaluated.

Reply:

Thank you for your suggestions. Here, we performed statistical analysis to obtain the yield of ZSM-5-MT crystals using three different SEM images. As shown in Fig. R9, the yield of ZSM-5-MT crystals is 42.7%, 41.2%, and 50.8% in Fig. R5a, b, and c, respectively. Thus, we can conclude that the yield of MT structures in ZSM-5-MT sample is among 40% and 50%.

Figure R9. The statistics of the yield of ZSM-5-MT crystals using SEM images. ZSM-5-MT crystals are marked by red dots. The yield of ZSM-5-MT crystals 42.7% (a), 41.2% (b), and 50.8% (c).

7. The MT structure is very unique and the growth mechanism should be investigated.

Reply:

Thank you for your suggestions. We fully agree with your comments and we have added relevant contents into the revised supplementary information.

To investigate the growth mechanism of ZSM-5-MT, we captured some embryonic ZSM-5-MT crystal without fully grown structure in the course of zeolite synthesis. In Fig. R10a-c, a darker region with different size can be observed at the interface section between the tenon and mortise subunits in each embryonic ZSM-5-MT crystal. Due to the fact that the growth rate along **a** axis is higher than that along the **b** axis, the tenon crystal grows into the mortise crystal to fill the gap between two crystals¹. Furthermore, we find some embryonic ZSM-5-MT crystals consisting of more than two components

(Fig. R10d-e), which indicates that discrete zeolite crystals can assemble into a primary ZSM-5-MT crystal. Then, this primary crystal was converted into the complete ZSM-5-MT structure during further crystallization.

As mentioned in Methods, the heating rate during crystallization was decreased and the amount of NaOH was increased to obtain the ZSM-5-MT crystals. Increasing alkalinity will speed up the crystallization of zeolites. And, we anticipate that slowing down the heating rate during crystallization can increase the amount of crystal nuclei and discrete zeolite crystals². As a result, the probability to form intergrowth structure increased with an increase of the density of crystal nuclei and discrete zeolite crystals. Based on these results and previous studies³, the schematics of the growth mechanism of ZSM-5-MT crystals is presented in Fig. R10g.

Figure R10. (a-f) ADF-STEM images to show the growth of the ZSM-5-MT crystals during the crystallization. (g) Schematics of the structural evolution of the ZSM-5-MT crystals.

New references were added in the revised supplementary information, which include:

[1] Karwacki L, *et al.* Morphology-dependent zeolite intergrowth structures leading to distinct internal and outer-surface molecular diffusion barriers. *Nature Materials* **8**, 959-965 (2009). [2] Xu R, Pang W, Yu J, Huo Q, Chen J. *Chemistry of Zeolites and Related Porous Materials: Synthesis and Structure*. (John Wiley & Sons, Singapore, 2007). [3] Dai W, *et al.* Platelike MFI Crystals with Controlled Crystal Faces Aspect

Ratio. *Journal of the American Chemical Society* **143**, 1993-2004 (2021).

8. The authors have claims such as "...atomic interface structure, line 75", "...tenon-mortise intergrowth in atomic precision, line 179", which seem to be overstated. The two components of the heterostructure differ in height and cannot be imaged with an atomically-resolved interface.

Reply:

Thank you for your suggestions. As we have mentioned in above comments, oxygen atom columns can be clearly resolved. However, the positions of Si atom columns are blurred due to the overlapping contrast. Thus, we deleted the "atomic" in "...atomic interface structure, line 75" and the "atomic precision" in "...tenon-mortise intergrowth in atomic precision, line 179".

9. The discussion about contents of Fig. 1g and h is confusing. From line 128, the lower part of Fig. 1g is in-focus with sinusoidal channels, which seems to contradicts with the labels in Fig. 1h.

Reply:

Thank you for your comments. First of all, we have to apologize for these confusing contents result from the wrong labels in Fig. 1h. We have added the correct labels to Fig. 1g and Fig. 1h, respectively.

References:

[1] Karwacki L, *et al.* Morphology-dependent zeolite intergrowth structures leading to distinct internal and outer-surface molecular diffusion barriers. *Nature Materials* **8**, 959-965 (2009).

[2] Xu R, Pang W, Yu J, Huo Q, Chen J. *Chemistry of Zeolites and Related Porous Materials: Synthesis and Structure*. (John Wiley & Sons, Singapore, 2007).

[3] Dai W, *et al.* Platelike MFI Crystals with Controlled Crystal Faces Aspect Ratio. *Journal of the American Chemical Society* **143**, 1993-2004 (2021).

Referee #2

We are very grateful to your professional comments on the acidity property and catalytic performance of the sample, although our focus is mainly on the interface structure in atomic scale. Actually, the creation of inherent framework-associated Al Lewis acid sites (LASs) is due to the oxygen vacancies at the interface by the lattice mismatch of zeolite intergrowth, rather than conventional extra-framework Al LASs. We improved the acidity characterization following your comments. The novel structure exhibited a lower density of LASs and BASs and an unchanged ratio of BASs/LASs. In addition, we validated the overall enhanced effect of our sample (ZSM-5-MT) on the catalytic performance, as following your suggestion to distinguish the hydrogen transfer reaction path in different stage of reaction, which is methanol-induced in the initial stage of the reaction and is olefins-induced in the sufficient reaction stage. Our sample exhibited a drastic effect in suppressing the yield of light alkanes in the sufficient stage of reaction, and an effect in increasing light alkanes in initial stage of reaction. Considering the product in the initial stage of reaction contributed insignificantly to the gross yield or gross selectivity of olefins in entire reaction process, our original statement is correct and sound. The supplemental data indeed provide deep understanding of the catalytic performance of our sample.

The quality of the manuscript is indeed improved significantly with your helpful suggestions, which will not only benefits our team, but also wide readers of the journal. We replied your comments point by point as follows.

Thanks again.

1. The acid properties of the zeolites used in this work are key to prove the authors' conclusions. For this, the use of NH_3 and pyridine adsorption is a routine analysis for zeolites and the tests are very standardized. Thus, I see several deficiencies on the results presentation and discussion when using these probe molecules to measure the zeolite acidity. Nowadays, it is easy to quantify the amount of adsorbed NH_3 and therefore the amount of desorbed NH_3 can be reliably quantified. I do not see the total

acidity measured by NH_3 adsorption. Besides, it is unacceptable to present NH_3 -TPD of two samples using arbitrary units (using arbitrary units could work for one sample). Additionally, the authors should refer to acid strength when analyzing the TPD profiles. Likewise, the FTIR spectra of pyridine-saturated zeolites are not properly presented, additional treatments can improve the result analysis, such including a deconvolution of the bands. At a glance, the BAS concentration in the ZSM-5-Sb zeolite looks to be lower than that in the ZSM-5-MT (inferred from the peak intensity/area). The authors should check this and improve the correlation of acid properties with the catalyst performance.

Reply:

Thank the referee for the comments. We added quantitative NH_3 -TPD profiles of two samples (Figure R1) in supplementary information. The arbitrary unit in NH_3 -TPD profiles was used for one sample. Meanwhile, we also give the total acid sites density of the two samples in Figure R1. The total acid sites density decreased over ZSM-5-MT compared to ZSM-5-Sb, implying that some acid sites were changed by the introduction of mortise-tenon interface.

Figure R1. NH_3 -TPD for ZSM-5-MT (a) and ZSM-5-Sb (b). The acid density was normalized to the weight of sample.

In addition, we have carefully checked and improved the FTIR spectra of adsorbed pyridine. We examined the FTIR spectra of adsorbed pyridine of ZSM-5-MT and ZSM-5-Sb catalysts again, and the improved results are given in Figure R2a. There is no need

to perform a deconvolution of the bands, as the characteristic bands corresponding to LAS and BAS are all single signals. The concentration of BAS and LAS were respectively quantified according to IR bands at 1545 cm^{-1} and 1455 cm^{-1} . As summarized in Table R1, the concentration of BAS in ZSM-5-Sb ($207.8\text{ }\mu\text{mol/g}$) is higher than that in ZSM-5-MT ($167.6\text{ }\mu\text{mol/g}$), and the concentration of LAS in ZSM-5-Sb ($17.6\text{ }\mu\text{mol/g}$) is slightly higher than that that in ZSM-5-MT ($14.2\text{ }\mu\text{mol/g}$), while their ratio of BASs/LASs are equal.

Considering the FTIR spectra of adsorbed pyridine are unable to distinguish the Al species of Lewis acid sites, we further detected these acid sites using CO as probe molecules for the FTIR spectroscopy and ^{27}Al magic-angle-spinning nuclear magnetic resonance (MAS-NMR). In Figure R2b, the peak at 2129 cm^{-1} indicates the physisorbed CO and the peak at 2168 cm^{-1} results from the CO interacting with the BASs in zeolites. Interestingly, the ZSM-5-MT shows two bands at $\sim 2338\text{ cm}^{-1}$ and 2358 cm^{-1} , which are attributed to the CO interactions with penta-coordinated and tri-coordinated LASs respectively after the dehydration of zeolites during sample pretreatment (Fig.R2c), while the ZSM-5-Sb does not. Such Al-CO interactions can be promoted by the complete or partial de-coordination of water from the fully hydrated tri-coordinated aluminum species. Therefore, appreciable amounts of the tri-coordinated aluminum sites can only be found in ZSM-5-MT zeolites. Combining with the FTIR spectra of adsorbed pyridine results, it can be concluded that some tetra-coordinated aluminum species transformed into tri-coordinated aluminum species due to the inherent missing of O atoms at the tenon-mortise interfaces, which agrees with the decrease on BAS concentration.

The ^{27}Al MAS NMR spectra (Fig.R2d) of both ZSM-5-MT and ZSM-5-Sb show sharp peaks at 58 ppm that are attributed to tetra-coordinated Al in bulk framework. It validates that most of Al atoms are incorporated into framework. Meanwhile, the ZSM-5-Sb shows a broad peak of chemical shift at 0 ppm belongs to distorted octahedral Al species, which is experimentally assigned to extra-framework Lewis acidic Al species¹. As for the ZSM-5-MT, the peak at 0 ppm disappeared, while a new peak at 4 ppm corresponding to the boehmite-like structures arose. In other words, there almost do not

exist extra-framework aluminum species in ZSM-5-MT. Meanwhile, such boehmite-like structures only exist when the high-density fully hydrated tri-coordinated LASs are very close in space to connect, for example, when such LASs are concentrated at the interface of zeolite intergrowth. Based on the FTIR and ^{27}Al MAS NMR results, the inherent tri-coordinated framework-associated Al (Al_{FR}) LASs caused by the missing of O atoms in ZSM-5-MT are confirmed experimentally.

The discussions on the correlation of acid properties with the catalyst performance are presented in question #2. We have added all these new experiments and discussions into the revised manuscript and supplementary information.

Figure R2. **a**, FTIR spectroscopy of adsorbed pyridine in ZSM-5-MT (black line) and ZSM-5-Sb (red line). **b**, FTIR spectroscopy of adsorbed CO in ZSM-5-MT (black line) and ZSM-5-Sb (red line). **c**, Structural evolution of tri-coordinated Al_{FR} LASs under different conditions. **d**, Al MAS NMR of ZSM-5-MT (black line) and ZSM-5-Sb (red line).

Table R1. The acid site concentrations of ZSM-5-MT and ZSM-5-Sb determined from FTIR spectra of adsorbed pyridine.

Sample	Acid site concentration ($\mu\text{mol/g}$)		ratio of BASs/LASs
	LAS	BAS	
ZSM-5-MT	14.22	167.63	11.79
ZSM-5-Sb	17.62	207.83	11.80

2. The catalyst tests are pretty simple and do not greatly support the authors' claims. To better observe kinetic differences among different catalysts, it is interesting to evaluate the conversion and product yield or selectivity at different space-times. You can find a good example of this in these publications: <https://doi.org/10.1021/jp4053677>, <https://doi.org/10.1021/jacs.6b09605>. Varying the contact time, varies the conversion and product yield/selectivity, and what is interesting is to compare the product yield/selectivity at different conversions (different from 100%). Thus, a catalyst would be kinetically different from other if the product yield/selectivity is different at the same conversion (conversion different from 100%). For this, it would be interesting to see curves of product yield/selectivity against conversion. These comparisons provide a better understanding of the catalyst performance differences. Obviously, a catalyst with a lower BAS density operating at 100% conversion would yield more light olefins than a catalyst with a higher BAS density because light olefins are intermediates in the methanol-to-hydrocarbons reaction. Furthermore, the authors should correlate and quantify the catalyst performance with the acid properties.

Reply:

Thank you for your comments and suggestions.

As indicated in the reference you suggested (<https://doi.org/10.1021/jacs.6b09605>), the olefin induced hydrogen transfer (OIHT) and methanol induced hydrogen transfer (MIHT) acting as the dominant hydrogen transfer route in sufficient reaction stage and initial stage of the reaction, respectively. Therefore, the catalytic performances of both two samples were respectively examined at full and partial methanol conversion. And, the catalytic performance and acid properties of zeolite catalysts were also correlated

following your comments. The detailed discussions are as followed.

The steady methanol conversion inside zeolite channels follows the hydrocarbon pool (HP) mechanism consisting of olefins-based and aromatics-based cycles. The olefin induced hydrogen transfer (OIHT) acting as the dominant step bridging the two cycles². Actually, the rate of OIHT is solely depend on the concentration of BAS³. Bearing this in mind, more vigorous OIHT will be observed over ZSM-5-Sb compared to ZSM-5-MT at total methanol conversion. To address the effect of decrease on BAS concentration, we firstly tested the catalytic performances of ZSM-5-MT and ZSM-5-Sb catalysts in methanol conversion at total conversion rate ($\geq 99\%$). The gas product selectivities are given in Fig. R3a-c, which indicates that two catalysts with different architectures exhibit quite different catalytic performances. The selectivities of propylene (25-29%) and butene (12-13%) over ZSM-5-MT are higher than those over ZSM-5-Sb (18-19% and $\sim 11\%$, respectively). Meanwhile, the ZSM-5-MT shows lower selectivities of C₁-C₅ alkanes (32-38%) and aromatics ($\sim 8\%$) than the ZSM-5-Sb (42-44% and $\sim 12\%$, respectively). It is known in the hydrocarbon pool (HP) mechanism that the aromatics-based cycle produces almost equal amounts of ethylene and propylene, while the olefins-based cycle will produce much more propylene than ethylene⁴⁻⁶. Therefore, the relative contribution of two cycles can be represented by the ratio of ([propylene]-[ethylene])/[ethylene] ((P-E)/E). As expected, the descriptor (P-E)/E ratio of ZSM-5-MT is much higher than that of ZSM-5-Sb (Fig. R3a and b), implying that the olefins-based cycle is more significantly enhanced in ZSM-5-MT. And, another criterion for determining the relative contribution of two cycles is the hydrogen transfer index (HTI)⁷. As shown in Fig. R3d, the C₄-HTI and C₅-HTI of ZSM-5-MT are lower than those of ZSM-5-Sb, indicating the less vigorous hydrogen transfer reactions in ZSM-5-MT. To explicitly illustrate the OIHT capacity, we further evaluated the catalytic performances of ZSM-5-MT and ZSM-5-Sb catalysts in propylene conversion (Fig. R3e). The ZSM-5-MT catalyst shows lower selectivities of both propane ($\sim 16\%$) and butane ($\sim 6\%$), which are the predominant HT products in propylene conversion, than the ZSM-5-Sb catalyst ($\sim 22\%$ and $\sim 7\%$ respectively). Thus, all these results demonstrate that the suppressed OIHT is induced in ZSM-5-MT, due

to the decrease on BAS concentration.

Figure R3. Selectivity of light olefins (ethylene, propylene and butene) and the (P-E)/E ratio for MTH over ZSM-5-Sb (a) and ZSM-5-MT (b) catalysts. Test conditions: complete methanol conversion ($\geq 99\%$), 475 °C, weight hourly space velocity (WHSV) of 5 h⁻¹). c, Selectivities of hydrocarbon products in MTH over ZSM-5-MT and ZSM-5-Sb catalysts at different times on stream (TOS). d, Hydrogen transfer index of C4 (C4-HTI, black) and C5 (C5-HTI, red) species as a function of time on steam for MTH over ZSM-5-MT (solid) and ZSM-5-Sb (hollow). e, Selectivity of alkanes in the propylene conversion over ZSM-5-Sb and ZSM-5-MT catalysts.

At partial methanol conversion, a majority of hydrogen transfer products are formed by methanol induced hydrogen transfer (MIHT), which involving LASs and BASs³. Therefore, their reaction behaviors and kinetics at induction period came into focus to investigate the role of modulated Al_{FR} LASs (Fig. R4). As indicated in Figure R4a, the induction period of ZSM-5-Sb sample is shorter than that of ZSM-5-MT sample, due to the higher BAS concentration in ZSM-5-Sb sample. At partial methanol conversions (<90%), more light alkanes and HT products (aromatics + C₁₋₄ alkanes) were formed over ZSM-5-MT (Fig. R4b and e), while more aromatics were formed over ZSM-5-Sb (Fig. R4d). In the MIHT route, the generation of aromatics is related to both BASs and LASs, while the light alkanes are formed over LASs. Considering the slightly higher LASs density of ZSM-5-Sb, we concluded that the Al_{FR} LASs showing higher capacity for MIHT compared to conventional Al_{EF} LASs. As a note in passing, the higher aromatics yield in ZSM-5-Sb mainly results from its significantly higher BASs density. The yield of propylene decreased after 80% methanol conversion for both ZSM-5-MT and ZSM-5-Sb catalysts (Fig. R4c), due to further conversion to aromatics. When full methanol conversion reached, olefin induced hydrogen transfer (OIHT) over BAS plays a dominant role in the formation of HT products³. Thus, the higher BAS density of ZSM-5-Sb results in lower yield of propylene and higher HT products compared to ZSM-5-MT at total methanol conversion.

In conclusion, the mortise-tenon interface in ZSM-5-MT results in the formation of framework-associated Al (Al_{FR}) Lewis acid sites as well as the decrease of BAS concentration, which definitely altering the catalytic performance.

Figure R4. Conversion as a function of contact time on ZSM-5-MT and ZSM-5-Sb (a). Yields of hydride transfer products (b), propylene (c), and aromatics (d) as a function of the conversion for ZSM-5-MT and ZSM-5-Sb samples. Light alkanes (e) yield as function of methanol conversion on two samples at partial conversion.

We have added all these new experiments and discussions into the revised manuscript and supplementary information. A new reference was added in the revised manuscript, which is: 41. Müller S, Liu Y, Kirchberger FM, Tonigold M, Sanchez-Sanchez M, Lercher JA. Hydrogen Transfer Pathways during Zeolite Catalyzed Methanol Conversion to Hydrocarbons. *Journal of the American Chemical Society* **138**, 15994-16003 (2016).

References:

[1] Bailleul S, *et al.* A Supramolecular View on the Cooperative Role of Brønsted and Lewis Acid Sites in Zeolites for Methanol Conversion. *Journal of the American*

Chemical Society **141**, 14823-14842 (2019).

[2] Yarulina I, Chowdhury AD, Meirer F, Weckhuysen BM, Gascon J. Recent trends and fundamental insights in the methanol-to-hydrocarbons process. *Nature Catalysis* **1**, 398-411 (2018).

[3] Müller S, Liu Y, Kirchberger FM, Tonigold M, Sanchez-Sanchez M, Lercher JA. Hydrogen Transfer Pathways during Zeolite Catalyzed Methanol Conversion to Hydrocarbons. *Journal of the American Chemical Society* **138**, 15994-16003 (2016).

[4] Bjørgen M, Joensen F, Lillerud K-P, Olsbye U, Svelle S. The mechanisms of ethene and propene formation from methanol over high silica H-ZSM-5 and H-beta. *Catalysis Today* **142**, 90-97 (2009).

[5] Sun X, *et al.* On reaction pathways in the conversion of methanol to hydrocarbons on HZSM-5. *Journal of Catalysis* **317**, 185-197 (2014).

[6] Wang S, *et al.* Polymethylbenzene or Alkene Cycle? Theoretical Study on Their Contribution to the Process of Methanol to Olefins over H-ZSM-5 Zeolite. *The Journal of Physical Chemistry C* **119**, 28482-28498 (2015).

[7] Wang S, *et al.* Relation of Catalytic Performance to the Aluminum Siting of Acidic Zeolites in the Conversion of Methanol to Olefins, Viewed via a Comparison between ZSM-5 and ZSM-11. *ACS Catalysis* **8**, 5485-5505 (2018).

Referee #3

We are very grateful to your professional comments on the acidic property and XRD information of our samples. As made further careful observation, simulation and histogram of sample, we provide sufficient data to validate our structure. The quality of the manuscript is indeed improved significantly with your helpful suggestions.

We replied your comments point by point as follows.

Thanks again.

1. As shown in Fig. 3a, if authors want to compare the peak intensity of IR bands, authors should calibrate the sample amount or the IR bands should be calibrated by Si-O band at near 1800 cm^{-1} . Otherwise, it is difficult to compare the IR band intensity.

Reply:

Thank you for your comments and suggestions. The curves of FTIR spectroscopy of adsorbed pyridine have been calibrated by the sample amount (Fig. R1). Meanwhile, both the specific concentration of acid sites (BAS and LAS) of ZSM-5-MT and ZSM-5-Sb have been also calibrated by the sample amount (as shown in Table R1), respectively.

Figure R1. The calibrated FTIR spectroscopy of adsorbed pyridine in ZSM-5-MT

(black line) and ZSM-5-Sb (red line).

Table R1. The acid site concentrations of ZSM-5-MT and ZSM-5-Sb determined from FTIR spectra of adsorbed pyridine.

Sample	Acid site concentration ($\mu\text{mol/g}$)		ratio of BASs/LASs
	LAS	BAS	
ZSM-5-MT	14.22	167.63	11.79
ZSM-5-Sb	17.62	207.83	11.80

2. The XRD patterns in the SI are very poor. Authors should offer high quality XRD patterns of the samples.

Reply:

Thank you for your suggestion. We have offered high quality XRD patterns of the samples in Fig. R2. And, we have added this figure to the revised supplementary information as Fig. S3. Both ZSM-5-MT and traditional short-b-axis ZSM-5 (ZSM-5-Sb) show pure MFI zeolite phase and high crystallinity.

Figure R2. XRD patterns for ZSM-5-MT and ZSM-5-Sb samples.

REVIEWER COMMENTS

Reviewer #1 (Remarks to the Author):

The authors have made considerable improvement on the manuscript. There are still some technical issues that should be properly addressed before publication.

1. The authors determine the orientation relationship of the tenon-mortise zeolite crystals from a STEM image along a specific projection, which is insufficient. Such an orientation relationship should be validated from another projection. Electron diffraction is a much easier way to determine the orientation relationship between the two crystals from multiple projections.

2. The measurement of oxygen vacancies by the noisy iDPC-STEM image is questionable and overstated.

1. The authors claimed based on the weak Bragg spot in the FFT to resolve 1.46 Å and individual oxygen columns. However, even one-pixel profile may include the contrast from neighboring Si columns, as the Si-O bond is just around 1.5-1.6 Å. Moreover, the oxygen columns are clearly distorted and disordered as shown in Fig. R6d, so it is not possible to properly profile oxygen columns by only one pixel. 2. The authors should discuss in the manuscript whether image filters are used or not. The filtering effects may introduce artefacts to the quite weak oxygen contrast when denoising. If the images are raw images without filtering, then apparently the contrast is strongly modulated by noise and makes the quantitative analysis of oxygen contrast unreliable. In general, it is suggested to remove the corresponding discussions on the quantitative analysis of oxygen vacancies.

3. The authors conducted a nice comparison between experimental and simulated iDPC-STEM images. The detailed simulation methods and associated publications should be mentioned in the manuscript.

Reviewer #2 (Remarks to the Author):

The authors have made some of the suggested corrections but even though the overall work does not meet the expected level for being published in this journal. There is a lack of novelty and the answers to my questions are still very basic and do not provide a sufficient contribution to the field of acid catalysis (zeolites) and nothing new for the extensively studied methanol-to-hydrocarbons (MTH) reaction. What is disconcerting is that there are better works related to the MTH reaction in the literature, where other researchers have studied this reaction in more detail providing better experimental data. Besides, this work does not provide relevant methodologies either. This work may be of interest for the readers of other catalysis-focused journals, such as Applied Catalysis A, ACS Catalysis, Catalysis Today, Microporous and Mesoporous Materials, Catalysis Letters, and so on.

IMPORTANT: Authors claim that LAS structure (the way these sites are generated) affect the catalytic performance and this is not sufficiently proved in their work. To prove their “conclusion”, they must have used catalysts with the same concentration of BAS and LAS, so that any change in the MTH reaction performance is exclusively linked to the different structure of LAS in both catalysts. Using a catalyst with a higher concentration of BAS unavoidably hinders the study of other properties of the acid

sites. This is because the BAS concentration strongly affects the MTH reaction performance and therefore they are actually studying the effect of two variables at the same time making difficult to prove that the LAS structure solely affects the MTH reaction performance. So that, this statement in their conclusions:

“Based on the catalytic test, the resulting AIFR LASs in ZSM-5-MT catalysts significantly decreased BAS concentration, inducing a higher propylene selectivity in steady MTH; the AIFR LASs showing higher capacity for MIHT compared to conventional AIEF LASs.”

is very difficult to believe it and it is unsupported by their experimental design and data.

These are other comments I have from the last version of the manuscript:

- The redaction is quite deficient: poor English grammar with several mistakes.
- The redaction of lines 271-279 is awkward, looking like unconnected bullet points of class notes. These introductory phrases are notes from other works (basics of the MTH reaction) and I see no description/relevance of the authors' results.
- Some sentences make no sense, like this one: “The olefin induced hydrogen transfer (OIHT) acting as the dominant step bridging the two cycles³.” What is the sentence structure? It looks like an incomplete clause.
- The most important (and real kinetic performance of the catalysts) is what authors present in Figure S18 of the Supplementary Information. This figure confirms that the kinetic is basically governed by the concentration of BAS: lower BAS concentration, slower kinetics (more space time is needed to reach similar conversion levels). However, closely knowing the MTH reaction performance, I can tell that the experimental data is not the best (it requires better selection of experiments, e.g. values of space time). Furthermore, it is shocking that the authors do not discuss more about the autocatalytic nature of this reaction, which is evidenced by the conversion profile (space time vs. conversion curves) and analysis of products (conversion vs. yield).
- Lines 300-301: “Thus, all these results demonstrate that the suppressed OIHT is induced in ZSM-5-MT, due to the decrease on BAS concentration.” This is a very well-known and expected behavior; it is fine to recall it but it is not an original conclusion from the present work.
- The authors' conclusions is already a known behavior for the MTH reaction on regards to the BAS/LAS concentration of acid sites. Their results are not impactful

Reviewer #3 (Remarks to the Author):

After the modifications, I think that this work should be accepted for the publication.

Response to review comments

Referee #1

We are very grateful to your professional comments on the structure details of our samples. As made further careful electron diffraction experiments, we provide sufficient data to validate our structure. The quality of the manuscript is indeed improved significantly with your helpful suggestions, which will not only benefits our team, but also wide readers of the journal.

We replied your comments point by point as follows.

Thanks again.

1. The authors determine the orientation relationship of the tenon-mortise zeolite crystals from a STEM image along a specific projection, which is insufficient. Such an orientation relationship should be validated from another projection. Electron diffraction is a much easier way to determine the orientation relationship between the two crystals from multiple projections.

Reply:

Thank the referee for the constructive comments. The electron diffraction patterns of ZSM-5-MT crystal were recorded from three different projections (Figure R1a-c). As shown in Figure R1d-f, the (010) and (100) surfaces of tenon subunit are perfectly connected with the (100) and (010) surfaces of mortise subunit, respectively. Simultaneously, the tenon subunit and mortise subunit share the common **c** axis. Thus, our SAED results confirm that the crystallographic axes **a** and **b** are rotated by 90° around common **c** in space to form the tenon and mortise, which is consistent with the conclusions of iDPC-STEM imaging.

This figure has been added in the revised version of supplementary information.

Figure R1. **a-c** The STEM images of ZSM-5-MT crystals along [100], [010] and [001] projections of tenon subunit, respectively. **d-f** show selected area electron diffraction (SAED) patterns correspond to the areas marked by blue circle in **a-c**, respectively.

2. The measurement of oxygen vacancies by the noisy iDPC-STEM image is questionable and overstated. 1. The authors claimed based on the weak Bragg spot in the FFT to resolve 1.46 Å and individual oxygen columns. However, even one-pixel profile may include the contrast from neighboring Si columns, as the Si-O bond is just around 1.5-1.6 Å. Moreover, the oxygen columns are clearly distorted and disordered

as shown in Fig. R6d, so it is not possible to properly profile oxygen columns by only one pixel. 2. The authors should discuss in the manuscript whether image filters are used or not. The filtering effects may introduce artefacts to the quite weak oxygen contrast when denoising. If the images are raw images without filtering, then apparently the contrast is strongly modulated by noise and makes the quantitative analysis of oxygen contrast unreliable. In general, it is suggested to remove the corresponding discussions on the quantitative analysis of oxygen vacancies.

Reply:

Thank you for your comments. We fully agree with your comments and suggestions. Low-pass gaussian blur was used as image filters in our manuscript. As you suggested, we have added “The experimental iDPC-STEM images were filtered by low-pass gaussian blur for noise smoothing.” in the Method section of revised manuscript. As exemplified in Figure R2c, the filtering shows no effects on both the intensities and positions of four bridge oxygen-atom columns, it just locally smooth the profiles. Thus, we think filtering will not affect our results.

Figure R2. **a**, The raw iDPC-STEM image. **b**, The filtered image from **a**. **c**, The corresponding intensity profiles of the same area marked by yellow frame in **a** and **b**.

Particularly, in the previous version of the manuscript, we used the profile of oxygen columns to perform the **semi-quantitative analysis** of oxygen vacancies and illustrate the presence of oxygen vacancies. Further, we added “**The normalized intensities of O peaks can semi-quantitatively reflect the quantity of atoms in these O atom columns.**” in the revised manuscript to emphasize our semi-quantitative analysis of oxygen vacancies.

3. The authors conducted a nice comparison between experimental and simulated iDPC-STEM images. The detailed simulation methods and associated publications should be mentioned in the manuscript.

Reply:

As you suggested, we have added detailed image simulation methods and associated publications in the methods section of the revised manuscript.

The iDPC-STEM image simulations were carried out on the basis of the multi-slice approach¹, which can be extended to support iDPC-STEM as explained in the previous works²⁻⁴. The parameters for simulations, such as convergence semi-angle (15 mrad), collection angle (4-22 mrad), aberration coefficients and applied dose, were adopted from our imaging experiments. During the simulations, four-sector ADF images were obtained from four segments of the four-quadrant detector spanning 4–22 mrad and then applied to generate the iDPC images using the approach described in the literature². The final iDPC-STEM images were convolved with a two-dimensional Gaussian function of 80-pm full-width at half maximum to account for finite probe size.

References:

- [1] Kirkland, E. J. *Advanced Computing in Electron Microscopy* 3rd edn (Springer, 2020).
- [2] Lazić, I., Bosch, E. G. T. & Lazar, S. Phase contrast STEM for thin samples: integrated differential phase contrast. *Ultramicroscopy* 160, 265–280 (2016).
- [3] Lazić, I. & Bosch, E. G. T. in *Advances in Imaging and Electron Physics* Vol. 199

(ed. Hawkes, P. W.) Ch. 3, 75–184 (Elsevier, 2017).

[4] Bosch, E. G. T. & Lazić, I. Analysis of HR-STEM theory for thin specimen. *Ultramicroscopy* 156, 59–72 (2015).

Referee #2

We are very grateful to your professional comments on the kinetic performances of the catalysts. As made further careful kinetic experiments, we provide sufficient data to validate our original statements. The content of “**Catalytic performances of ZSM-5-MT catalyst**” in revised manuscript has been rewritten based on our improved kinetic experiments. The quality of the manuscript is indeed improved significantly with your helpful suggestions, which will not only benefits our team, but also wide readers of the journal.

We replied your comments point by point as follows.

Thanks again.

1. IMPORTANT: Authors claim that LAS structure (the way these sites are generated) affect the catalytic performance and this is not sufficiently proved in their work. To prove their “conclusion”, they must have used catalysts with the same concentration of BAS and LAS, so that any change in the MTH reaction performance is exclusively linked to the different structure of LAS in both catalysts. Using a catalyst with a higher concentration of BAS unavoidably hinders the study of other properties of the acid sites. This is because the BAS concentration strongly affects the MTH reaction performance and therefore they are actually studying the effect of two variables at the same time making difficult to prove that the LAS structure solely affects the MTH reaction performance. So that, this statement in their conclusions: “Based on the catalytic test, the resulting Al_{FR} LASs in ZSM-5-MT catalysts significantly decreased BAS concentration, inducing a higher propylene selectivity in steady MTH; the Al_{FR} LASs showing higher capacity for MIHT compared to conventional Al_{EF} LASs.” is very difficult to believe it and it is unsupported by their experimental design and data.

- The most important (and real kinetic performance of the catalysts) is what authors present in Figure S18 of the Supplementary Information. This figure confirms that the kinetic is basically governed by the concentration of BAS: lower BAS concentration, slower kinetics (more space time is needed to reach similar conversion levels). However, closely knowing the MTH reaction performance, I can tell that the experimental data is

not the best (it requires better selection of experiments, e.g. values of space time). Furthermore, it is shocking that the authors do not discuss more about the autocatalytic nature of this reaction, which is evidenced by the conversion profile (space time vs. conversion curves) and analysis of products (conversion vs. yield).

- Lines 300-301: “Thus, all these results demonstrate that the suppressed OIHT is induced in ZSM-5-MT, due to the decrease on BAS concentration.” This is a very well-known and expected behavior; it is fine to recall it but it is not an original conclusion from the present work.

- The authors’ conclusions is already a known behavior for the MTH reaction on regards to the BAS/LAS concentration of acid sites. Their results are not impactful.

Reply:

Thank the referee for the suggestions. In order to use catalysts with the same concentration of BAS and LAS for comparison, we synthesized a single-crystal ZSM-5 catalyst with an increased Si/Al ratio (~75), and the obtained sample is denoted as ZSM-5-Sb-75.

Figure R1. FTIR spectroscopy of adsorbed pyridine in ZSM-5-MT and ZSM-5-Sb-75.

Table R1. Supplementary Table 2. The acid site concentrations determined from FT-IR spectra of adsorbed pyridine.

Sample	Acid site concentration ($\mu\text{mol/g}$)		ratio of BASs/LASs
	LAS	BAS	
ZSM-5-MT	14.22	167.63	11.79
ZSM-5-Sb-67	17.62	207.83	11.80
ZSM-5-Sb-75	14.80	143.30	9.68

As summarized in Fig. R1 and Table R1, the BAS and LAS concentrations of ZSM-5-Sb-75 are nearly equal to these of ZSM-5-MT, respectively. Therefore, we tested the catalytic performances of ZSM-5-MT and ZSM-5-Sb-75 catalysts in methanol conversion at partial and total conversion rates.

In detail, the initial stage of the reaction is determined as the conversion of methanol is lower than 70-80%, considering the easy activation and easy reaction of methanol with zeolite. After that, it belongs to the sufficient stage of the reaction. More light alkanes (C_{1-4} alkanes) were formed over ZSM-5-MT compared with ZSM-5-Sb-75 (Fig. R2a), while aromatics in nearly the same amount were formed over two catalysts (Fig. R2b) when the conversion of methanol is lower than 75%. During this stage, a majority of hydrogen transfer products are formed by methanol induced hydrogen transfer (MIHT). The generation of aromatics is related to both BASs and LASs, while the light alkanes are formed over LASs. Considering their nearly identical acid sites concentrations, we concluded that the Al_{FR} LASs exhibit higher capacity for MIHT compared to conventional Al_{EF} LASs.

At the sufficient stage of reaction, the selectivities of propylene (44.3 %) and butene (16.3 %) over ZSM-5-MT are higher than those over ZSM-5-Sb-75 (35.8 % and ~ 8 %, respectively) (Fig. R2c-e). Meanwhile, the ZSM-5-MT shows lower selectivities of ethene (10.2 %) and aromatics (11.5 %) than the ZSM-5-Sb (16.6 % and ~ 19.6 %, respectively). As mentioned above, the olefins-based and the aromatics-based cycles existed in the steady conversion of methanol inside zeolite channels, following the HP

mechanism. Almost equal amounts of ethene and propylene were produced in the aromatics-based cycle, while the olefins-based cycle favored the production of much higher yield of propylene than ethene. Apparently, the intergrown structure of ZSM-5-MT exhibits the space confinement effect on inhibiting the aromatics-based cycle, while amplifying the olefins-based cycle. In addition, the higher ratio of propylene to propane and the higher ratio of butene to butane were retained with ZSM-5-MT as compared to ZSM-5-Sb. These further confirm that the Al_{FR} LASs of ZSM-5-MT suppress the hydrogen transfer effect significantly.

Figure R2. Yields of light alkanes (a), aromatics (b), propylene (c) and ethene (d) as a function of the conversion for ZSM-5-MT and ZSM-5-Sb-75 catalysts. Black square: ZSM-5-MT (B 167.6, L 14.2), red circle: ZSM-5-Sb-75 (B 143.3, L 14.8). e, Distribution of different carbon numbers in products for ZSM-5-MT and ZSM-5-Sb-75 catalysts. Test conditions: complete methanol conversion ($\geq 99\%$), 475 °C, weight hourly space velocity (WHSV) of 3 h⁻¹.

Noted here that the present sample contained only half mortise-tenon architectures (see Supplementary Fig. 3 in revised SI), which shows a high propylene selectivity of 44.3% that is comparable to state-of-the-art catalysts in MTO reactions (Table R2). As shown in Table R2, all the reported ZSM-5 based catalysts with extremely high propylene selectivity ($>40\%$) must be prepared by post-treatment, such as metal incorporation and dealumination. But, the ZSM-5-MT catalyst in our work was prepared via a one-pot

process. It can be speculated that the ZSM-5 sample with pure mortise-tenon architecture will be favorable to produce propylene and butene with much higher selectivity.

Table R2. Comparison of the catalytic performance of reported ZSM-5 based catalysts for the conversion of methanol to olefins.

Catalyst	Reaction temperature (°C)	Conversion (%)	Selectivity of C ₃ H ₆ (%)	Reference
H-ZSM-5-MT(Si/Al=67)	475	99.5	44.3	This work
TaAlS-1(0.013/0.027/1)	400	100	53	1
CaZSM-5_AE3	500	100	38	2
MgZSM-5_AE7	500	100	39	2
CaZSM-5_AE5	500	100	51	2
SrZSM-5_AE6	500	100	42	2
ZSM-5_Z1	500	100	30	2
Mesoporous ZSM-5_M1	500	100	38	2
Dealumination ZSM-5_M4	500	100	46	2
C-ZSM-5(Si/Al=400)	450	99.8	38.7	3

References:

1. Lin L, *et al.* Control of zeolite microenvironment for propene synthesis from methanol. *Nature Communications* **12**, 822 (2021).
2. Yarulina I, *et al.* Structure–performance descriptors and the role of Lewis acidity in the methanol-to-propylene process. *Nature Chemistry* **10**, 804-812 (2018).
3. Hu S, *et al.* Selective formation of propylene from methanol over high-silica nanosheets of MFI zeolite. *Applied Catalysis A: General* **445-446**, 215-220 (2012).

We have added Figure R2 as Figure 4 in our revised manuscript. And, these new discussions have been added into the revised manuscript.

2. These are other comments I have from the last version of the manuscript:

- The redaction is quite deficient: poor English grammar with several mistakes.
- The redaction of lines 271-279 is awkward, looking like unconnected bullet points of class notes. These introductory phrases are notes from other works (basics of the MTH reaction) and I see no description/relevance of the authors' results.
- Some sentences make no sense, like this one: "The olefin induced hydrogen transfer (OIHT) acting as the dominant step bridging the two cycles³." What is the sentence structure? It looks like an incomplete clause.

Reply:

Thank you for your suggestion. As you can see in question #1, we have rewritten the content of "**Catalytic performances of ZSM-5-MT catalyst**" in revised manuscript based on our improved kinetic experiments. And, the lines 271-279 have been deleted.

REVIEWER COMMENTS

Reviewer #1 (Remarks to the Author):

The authors have followed my suggestions and provided extra experimental evidences for their conclusions. This paper can now be considered for publication in Nature Communications. I have two additional comments that should be considered for next revision.

1. The authors provide electron diffractions along different zone axes of the heterostructure, which is nice. However, the electron diffractions are not properly explained. The a/b lattice constants are so close for ZSM-5 that it is totally impossible to distinguish between the mortise and the 90 degree rotated tenon simply by labelling the (200)/(020) spots. The authors should discuss this point based on the projection symmetry as well as the extinction rules in the main text, which is rather important. From the diffraction patterns, all spots are very sharp and no splitting or broadening of spots are observed. It indicates the mortise-tenon are perfectly registered. It is suggested to conduct a simple strain analysis across the interface to validate this point.
2. There are previous studies that also observed the 90 degree rotational boundaries for the ZSM-5 zeolite and these papers should be properly cited.

Reviewer #2 (Remarks to the Author):

The work of Wang et al. reports a great effort to synthesize a stilted ZSM-5 architecture based on the mortise and tenon terms from woodworkers. What authors really obtained is the tenon joint and not the mortise one. Hence, conceptually, the zeolite crystal architecture they obtained is just a tenon-joint structure.

However, what is more important and worrisome about this work is the lack of impact and novelty in the application (the methanol-to-olefins/hydrocarbons reaction). The authors have made a great effort to improve the kinetic tests, but the results reveal insignificant impact on the kinetics. First, looking at Fig. 4, there are some points that are out of the trend suggesting that the experiment should be verified (e.g. the red point at 80% conversion). Apart from that point, all the data obtained from both catalysts is quite similar in the whole range of conversions meaning no significant changes. Additionally, it is difficult to read Fig. 4 a-d because there is not a legend for the red and black data, and the conditions of the tests in the figure title should be revised (the results do not correspond to full conversion and therefore to a constant space velocity).

In summary, upon looking at the new kinetic data, it is clear that the great effort to synthesize an uncommon zeolite crystal architecture does not yield a great benefit to the methanol-to-hydrocarbons reaction. This is also evident and it should have been predictable when looking at the characterization of Bronsted and Lewis acid sites (Fig. 3a), with truly no impactful changes. Thus, my decision as an expert in

the area is to reject this manuscript for publication in Nature Communications for the abovementioned reasons.

Response to review comments

Referee #1

We are very grateful for your professional comments. As made further strain analysis experiments, we confirm that the mortise-tenon are perfectly registered. The quality of the manuscript is indeed improved significantly with your helpful suggestions.

We replied to your comments point by point as follows.

Thanks again.

1. The authors provide electron diffractions along different zone axes of the heterostructure, which is nice. However, the electron diffractions are not properly explained. The *a/b* lattice constants are so close for ZSM-5 that it is totally impossible to distinguish between the mortise and the 90 degree rotated tenon simply by labelling the (200)/(020) spots. The authors should discuss this point based on the projection symmetry as well as the extinction rules in the main text, which is rather important. From the diffraction patterns, all spots are very sharp and no splitting or broadening of spots are observed. It indicates the mortise-tenon are perfectly registered. It is suggested to conduct a simple strain analysis across the interface to validate this point.

Reply:

Thank the referee for the constructive comments. We carried out the strain analysis in two typical iDPC-STEM images from the [001] direction, since the mortise and tenon subunits have coincident *c*-axes and exhibit the smallest differences in height from this direction. In the GPA strain maps, there are no obvious strains that exist at the mortise-tenon interfaces, which indicates the mortise-tenon are perfectly registered.

This figure has been added in the revised version of supplementary information.

In addition, we added “Although the two subunits can not be distinguished in the electron diffraction patterns due to the similar lattice constants of **a** and **b** for ZSM-5 (Supplementary Fig.6), the sharp spots without splitting or broadening suggest that mortise and tenon are perfectly registered, which is further confirmed by strain analysis

across the interface (Supplementary Fig.7).” in the revised manuscript.

Figure R1. **a** and **c**, The iDPC-STEM images of ZSM-5-MT crystals from [001] direction. **b** and **d**, The GPA strain distribution maps corresponding to **a** and **c**, respectively. The interfaces are marked by red dotted frames in images.

2. There are previous studies that also observed the 90 degree rotational boundaries for the ZSM-5 zeolite and these papers should be properly cited.

Reply:

Thank the referee for the comments. In our current version, we have cited some references about the 90 degree rotational boundaries for the ZSM-5 zeolite. In the revised version, we added three more references in the revised manuscript.

The added references are:

36. Weidenthaler C, Fischer RX, Shannon RD, Medenbach O. Optical Investigations of Intergrowth Effects in the Zeolite Catalysts ZSM-5 and ZSM-8. *The Journal of Physical Chemistry* **98**, 12687-12694 (1994).

37. Roeffaers MBJ, *et al.* Morphology of Large ZSM-5 Crystals Unraveled by Fluorescence Microscopy. *Journal of the American Chemical Society* **130**, 5763-5772 (2008).

38. Kocirik M, Kornatowski J, Masařík V, Novák P, Zikánová A, Maixner J. Investigation of sorption and transport of sorbate molecules in crystals of MFI structure

type by iodine indicator technique. *Microporous and Mesoporous Materials* **23**, 295-308 (1998).

Referee #2

1. The work of Wang et al. reports a great effort to synthesize a stilted ZSM-5 architecture based on the mortise and tenon terms from woodworkers. What authors really obtained is the tenon joint and not the mortise one. Hence, conceptually, the zeolite crystal architecture they obtained is just a tenon-joint structure.

Reply:

Thank the referee for your affirmation of our great effort to synthesize a mortise-tenon ZSM-5 architecture. The design of this novel ZSM-5 architecture is inspired by Chinese ancient timber buildings. A mortise-tenon joint is an interlocking of two pieces, where a tenon tongue on the end of a rail fits into a mortise hole on the side of another piece (Figure R1). Our careful characterizations demonstrated the tenon crystal grows into the mortise crystal to form a perpendicular intergrowth structure in ZSM-5-MT, which is just like the structure of a traditional mortise-tenon junction.

Figure R1. Schematic of a mortise-tenon joint.

2. However, what is more important and worrisome about this work is the lack of impact and novelty in the application (the methanol-to-olefins/hydrocarbons reaction). The authors have made a great effort to improve the kinetic tests, but the results reveal insignificant impact on the kinetics. First, looking at Fig. 4, there are some points that are out of the trend suggesting that the experiment should be verified (e.g. the red point at 80% conversion). Apart from that point, all the data obtained from both catalysts is quite similar in the whole range of conversions meaning no significant changes. Additionally, it is difficult to read Fig. 4 a-d because there is not a legend for the red and black data, and the conditions of the tests in the figure title should be revised (the results do not correspond to full conversion and therefore to a constant space velocity). In summary, upon looking at the new kinetic data, it is clear that the great effort to synthesize an uncommon zeolite crystal architecture does not yield a great benefit to the methanol-to-hydrocarbons reaction. This is also evident and it should have been predictable when looking at the characterization of Bronsted and Lewis acid sites (Fig. 3a), with truly no impactful changes. Thus, my decision as an expert in the area is to reject this manuscript for publication in Nature Communications for the abovementioned reasons.

Reply:

Thank the referee for your comments on the figure format. Three sets of repeated experiments are carried out on each sample under the same conditions to estimate the error and repeatability. We have added error bars and legends in Fig. 4 to rule out your concerns (see Figure R2). In addition, the conditions of the tests in the figure title are referred to the Fig. 4e, which shows the distribution of different carbon numbers of hydrocarbon products at complete methanol conversion. To rule out your confusion, this figure title has been revised.

Our kinetic experiments reveal that Al_{FR} LAS-enriched mortise-tenon interface shows a significant impact on the kinetics. At the initial stage of reaction, more light alkanes (C₁₋₄ alkanes) were formed over ZSM-5-MT compared with ZSM-5-Sb-75 (Fig. R2a), while aromatics in nearly the same amount were formed over two catalysts (Fig. R2b)

when the conversion of methanol is lower than 75%. During this stage, a majority of hydrogen transfer products are formed by methanol induced hydrogen transfer (MIHT). The generation of aromatics is related to both BASs and LASs, while the light alkanes are formed over LASs. Considering their nearly identical acid sites concentrations, we concluded that the Al_{FR} LASs exhibit higher capacity for MIHT compared to conventional Al_{EF} LASs.

At the sufficient stage of reaction (conversion > 80%), nearly equal light alkanes (C_{1-4} alkanes) were formed over ZSM-5-MT and ZSM-5-Sb-75 (Fig. R2a), while more aromatics were formed over ZSM-5-Sb-75 compared with ZSM-5-MT.

At complete methanol conversion ($\geq 99\%$), the selectivities of propylene (44.3 %) and butene (16.3 %) over ZSM-5-MT are higher than those over ZSM-5-Sb-75 (35.8 % and ~ 8 %, respectively) (Fig. R2c-e). Meanwhile, the ZSM-5-MT shows lower selectivities of ethene (10.2 %) and aromatics (11.5 %) than the ZSM-5-Sb (16.6 % and ~ 19.6 %, respectively). In addition, the higher ratio of propylene to propane and the higher ratio of butene to butane were retained with ZSM-5-MT as compared to ZSM-5-Sb.

Figure R2. **Catalytic performance of ZSM-5-MT catalyst.** Yields of light alkanes (a), aromatics (b), propylene (c) and ethene (d) as a function of the conversion of methanol

with ZSM-5-MT and ZSM-5-Sb-75 catalysts. **e**, Distribution of different carbon numbers of hydrocarbon products for ZSM-5-MT and ZSM-5-Sb-75 catalysts at complete methanol conversion ($\geq 99\%$) (Test conditions: 475 °C, weight hourly space velocity (WHSV) of 3 h⁻¹).

We have added Figure R2 as Figure 4 in our revised manuscript.

In this work, we use the lattice mismatch of zeolite intergrowth to create a framework-associated Al (Al_{FR}) Lewis acids (LAS)-enriched interface in a mortise-tenon ZSM-5 catalyst (ZSM-5-MT) with 90° intergrowth structures. Based on state-of-the-art electron microscopy, the atomic lattices and interface structures of the ZSM-5-MT catalyst were resolved with atomic precision. More importantly, the O atoms can be clearly identified with ultra-high resolution and a high signal-to-noise ratio. We find that the missing O atoms at this interface contribute to the formation of Al_{FR} LASs, which is different from conventional extra-framework Al LASs. The enrichment of Al_{FR} LASs in ZSM-5-MT was confirmed by using various methods, including the ²⁷Al solid state nuclear magnetic resonance (NMR) and Fourier transform infrared (FTIR) spectroscopy of adsorbed probe molecules (carbon monoxide and pyridine). It should be noted that although the concentrations of LAS of ZSM-5-MT and ZSM-5-Sb zeolite are nearly equal, their Lewis acidic Al species are completely different.

This work exhibits the following breakthroughs in catalysis and material science. First, we expand the material design concept of interface engineering to the field of porous materials. Then, based on the interface engineering, we also provide a general method to design the framework-associated Al (Al_{FR}) LASs in zeolites for tailored catalytic functions. Meanwhile, these results bridge the gap between the local structure and catalytic performance of Al_{FR} LASs at intergrowth interfaces.

We still need to emphasize that the present sample contained only half mortise-tenon architectures (see Supplementary Fig. 3 in revised SI), which shows a high propylene selectivity of 44.3% that is comparable to state-of-the-art catalysts in MTO reactions (Table R2). As shown in Table R2, all the reported ZSM-5 based catalysts with extremely high propylene selectivity (>40%) must be prepared by post-treatment, such

as metal incorporation and dealumination. But, the ZSM-5-MT catalyst in our work was prepared via a one-pot process. It can be speculated that the ZSM-5 sample with pure mortise-tenon architecture will be favorable to produce propylene and butene with much higher selectivity.

Thus, we believe that this work brings a new opportunity for the sustainable development of the ZSM-5 zeolite family and the methanol-to-olefins/hydrocarbons reaction, which is of general interest for a broad readership of *Nature Communications*.

Table R2. Comparison of the catalytic performance of reported ZSM-5 based catalysts for the conversion of methanol to olefins.

Catalyst	Reaction temperature (°C)	Conversion (%)	Selectivity of C ₃ H ₆ (%)	Reference
H-ZSM-5-MT(Si/Al=67)	475	99.5	44.3	This work
TaAIS-1(0.013/0.027/1)	400	100	53	1
CaZSM-5_AE3	500	100	38	2
MgZSM-5_AE7	500	100	39	2
CaZSM-5_AE5	500	100	51	2
SrZSM-5_AE6	500	100	42	2
ZSM-5_Z1	500	100	30	2
Mesoporous ZSM-5_M1	500	100	38	2
Dealumination ZSM-5_M4	500	100	46	2
C-ZSM-5(Si/Al=400)	450	99.8	38.7	3

References:

1. Lin L, *et al.* Control of zeolite microenvironment for propene synthesis from methanol. *Nature Communications* **12**, 822 (2021).
2. Yarulina I, *et al.* Structure–performance descriptors and the role of Lewis acidity in the methanol-to-propylene process. *Nature Chemistry* **10**, 804-812 (2018).
3. Hu S, *et al.* Selective formation of propylene from methanol over high-silica

nanosheets of MFI zeolite. *Applied Catalysis A: General* **445-446**, 215-220 (2012).

REVIEWER COMMENTS

Reviewer #2 (Remarks to the Author):

The work of Wang et al. does not sufficiently demonstrate that the two catalysts they synthesized have different kinetic behavior. Thus, this work is not suitable for publication in the prestigious journal of Nature Communications for this and these reasons:

The conversion-yield curves are not characteristic for the MTH/MTO/MTP/MTG reaction on ZSM-5 catalysts (see Fig. 23 in <https://doi.org/10.1039/C5CS00304K>). This invites to revise the experimental data and/or to check the experimental setup. It is well-known that the light olefins yield reaches a maximum and then decreases at high conversions (above 95%) on ZSM-5 catalysts, which is well-explained by the fact that light olefins are intermediates. The decreasing behavior of light olefins is further observed with an excess of catalyst (at full conversion regimes). Thus, the highest selectivity is observed at low conversions (when intermediates are formed and barely continues reacting to form final products such as aromatics and light paraffins). This explanation is the same for representation of the yield or selectivity data against space time/velocity (Fig. 2 in <https://doi.org/10.1080/10916466.2018.1555589>, Fig. 1 in <https://doi.org/10.1039/C7CY00129K>), a behavior early described by the pioneers of the MTH reaction Silvestri and Chang ([https://doi.org/10.1016/0021-9517\(77\)90172-5](https://doi.org/10.1016/0021-9517(77)90172-5)).

It is unclear that the ZSM-5-MT catalyst is better than others reported in the literature. By just tuning the Si/Al ratio (without further post-synthesis modifications), propylene selectivity can be as high as 45.9% (see [https://doi.org/10.1016/S1003-9953\(10\)60198-3](https://doi.org/10.1016/S1003-9953(10)60198-3)). Tuning the Si/Al is perhaps the easiest way to synthesize a catalyst selective to light olefins (mostly propylene). It is expected that high Si/Al ratios decrease the acid site density and therefore favor the light olefins formation and improve the catalyst stability (an issue that the authors did not study).

Besides, the propylene selectivity as an indicator to justify the catalyst performance is not well supported throughout the manuscript, which debilitates the message. The authors wrote some basis of the MTH reaction in the introduction, without mentioning the importance/interest of making propylene or even the existence of a technology called "MTP" or "MTO", the typical catalysts and the best ones. Their real objective is unclear.

This brings me to advise that, from an engineering viewpoint, the use of ZSM-5 catalysts is not practical to produce olefins from methanol (MTO technology). The current extended processes based on the MTO reaction use SAPO-34 catalysts that yield more than 80% of light olefins. The MTP technology certainly uses catalysts based on ZSM-5 zeolites, but this technology is not as extended or interesting as the MTO one.

The authors literally said "Thus, we believe that this work brings a new opportunity for the sustainable development of the ZSM-5 zeolite family and the methanol-to-olefins/hydrocarbons reaction"

Sustainable? The synthesis method they used is pretty conventional. The method involves the use of several environmentally unfriendly reagents and generates a significant amount of contaminated water. Besides, the authors failed to explain how to obtain the structure they claim (mortise-tenon). The only difference between both syntheses is the temperature ramp in the crystallization step (autoclave). Is that enough to obtain crystals with the claimed structure? How is the claimed architecture of the crystals truly obtained?

Finally, I must say that I truly acknowledge the enormous effort of the authors to improve their manuscript. Their work could be of interest for the readerships of a journal more focused on zeolite materials.

Reviewer #4 (Remarks to the Author):

I have now read for the first time the article of Wang et al., which is a revised version of an earlier submission, together with the rebuttal letter on two initial referees. Hence, I am providing totally new referee comments as I was not involved in the first round of refereeing of this article.

The paper reports on a detailed characterization of a so-called mortise-tenon ZSM-5 inter growth by using iDPC-STEM, which was used by the same group to investigate in detail MOFs materials, as reported in references 31 and 32, one article which also reached the same journal the authors now intend to publish their work. The authors show that it is possible to image missing O atoms due to the lattice mismatch in the inter growth, and hence induces the formation of AIFR Lewis acid sites, which in the opinion of the authors are responsible for the observed promotion of the olefinic cycle; and hence results in more propylene and butene formation. While the STEM data are spatially resolved, the other methods used are bulk methods, focusing on the state of Al and acidity, as evaluated with NMR and IR with probe molecules. The novelty of the article is in my opinion in the use of the iDPC-STEM image to investigate in great detail an inter growth structure of an industrially relevant zeolite, such as ZSM-5, while the other aspects are in my opinion important but less developed, worked. As such the work deserves publication in a high-impact journal, but in its current version I cannot recommend the work for publication.

First of all, I wish to stress that intergrowth structures can be present in different forms and shapes for zeolite ZSM-5 as shown in the 2009 Nature Materials of Karwacki, Kox and co-workers. While the works of a.o. Stavitski et al., Kox et al., and Roeffaers et al. report mainly the classical coffin-shaped intergrowth structure, interestingly there are reports out in which the "mortise-tenon" intergrowth structure has been shown to exist; an example can be found in the paper of Kox et al., Chem. Eur. J. 2008, 14, 1718-1725. It would be evaluate the proposed structure with the one described in this article; furthermore, one can also find examples, although not exactly the same in the paper of Mores et al., Chem. Eur. J. 2011, 17, 2874-2884. What is important to realise is that the "woodworkers" concept is not that straightforward as proposed by the authors as the total penetration should not always be the case, nor can it be guaranteed that there is no Al zoning; and as I also expect that there is at the interface

disorders/defects/missing atoms, hence Lewis acid sites, the bulk methods taken to evaluate this in the work are not fully conclusive. For this, the authors should have been used spatially resolved methods, although their crystals studied are smaller than those described in the above-mentioned references. Nevertheless, how sure are we that the differences are related to changes in LAS and not in changes in pore accessibility, number of BAS, as well as their strength. These points have to be addressed in a revised article before the paper can be published.

Response to review comments

Referee #2

1. The conversion-yield curves are not characteristic for the MTH/MTO/MTP/MTG reaction on ZSM-5 catalysts (see Fig. 23 in <https://doi.org/10.1039/C5CS00304K>). This invites to revise the experimental data and/or to check the experimental setup. It is well-known that the light olefins yield reaches a maximum and then decreases at high conversions (above 95%) on ZSM-5 catalysts, which is well-explained by the fact that light olefins are intermediates. The decreasing behavior of light olefins is further observed with an excess of catalyst (at full conversion regimes). Thus, the highest selectivity is observed at low conversions (when intermediates are formed and barely continues reacting to form final products such as aromatics and light paraffins). This explanation is the same for representation of the yield or selectivity data against space time/velocity (Fig. 2 in <https://doi.org/10.1080/10916466.2018.1555589>, Fig. 1 in <https://doi.org/10.1039/C7CY00129K>), a behavior early described by the pioneers of the MTH reaction Silvestri and Chang ([https://doi.org/10.1016/0021-9517\(77\)90172-5](https://doi.org/10.1016/0021-9517(77)90172-5)).

Reply:

Actually, it is correct that the light olefins yield reaches a maximum at high conversions, as light olefins are intermediates. But, the decrease of light olefins yield is not necessary. As shown in Fig. R1, the conversion-propylene yield curves in *Ref. 44* (Lercher et al, *Journal of the American Chemical Society* **138**, 15994-16003 (2016), still increases at high conversions (above 95%) but with a lower increase rate. Apparent, the slowdown of the increase rate validated the transformation of propylene intermediates.

Fig. R1. The conversion-propylene yield curves. The weight hourly space times correspond to the conversion rates.

2. It is unclear that the ZSM-5-MT catalyst is better than others reported in the literature. By just tuning the Si/Al ratio (without further post-synthesis modifications), propylene selectivity can be as high as 45.9% (see [https://doi.org/10.1016/S1003-9953\(10\)60198-3](https://doi.org/10.1016/S1003-9953(10)60198-3)). Tuning the Si/Al is perhaps the easiest way to synthesize a catalyst selective to light olefins (mostly propylene). It is expected that high Si/Al ratios decreases the acid site density and therefore favors the light olefins formation and improve the catalyst stability (an issue that the authors did not study). Besides, the propylene selectivity as an indicator to justify the catalyst performance is not well supported throughout the manuscript, which debilitates the message. The authors wrote some basis of the MTH reaction in the

introduction, without mentioning the importance/interest of making propylene or even the existence of a technology called “MTP” or “MTO”, the typical catalysts and the best ones. Their real objective is unclear. This brings me to advise that, from an engineering viewpoint, the use of ZSM-5 catalysts is not practical to produce olefins from methanol (MTO technology). The current extended processes based on the MTO reaction use SAPO-34 catalysts that yield more than 80% of light olefins. The MTP technology certainly uses catalysts based on ZSM-5 zeolites, but this technology is not as extended or interesting as the MTO one.

Reply:

The keys of our work are atomically resolving the local intergrowth structures and revealing the structure-properties relationship.

3. The authors literally said “Thus, we believe that this work brings a new opportunity for the sustainable development of the ZSM-5 zeolite family and the methanol-to-olefins/hydrocarbons reaction” Sustainable? The synthesis method they used is pretty conventional. The method involves the use of several environmentally unfriendly reagents and generates a significant amount of contaminated water. Besides, the authors failed to explain how to obtain the structure they claim (mortise-tenon). The only difference between both syntheses is the temperature ramp in the crystallization step (autoclave). Is that enough to obtain crystals with the claimed structure? How is the claimed architecture of the crystals truly obtained?

Reply:

The comment “The method involves the use of several environmentally unfriendly reagents and generates a significant amount of contaminated water” is irrelevant to the focus of our manuscript. The comments of the synthesis method are also irrelevant to the focus of our manuscript.

Referee #4

We are very grateful for your recommendation of our manuscript and your professional comments. Based on your comments, we have made a revision on the manuscript and answered all your questions as follows. After this revision, we sincerely ask you to reconsider our revised manuscript for further publication.

1. The paper reports on a detailed characterization of a so-called mortise-tenon ZSM-5 inter growth by using iDPC-STEM, which was used by the same group to investigate in detail MOFs materials, as reported in references 31 and 32, one article which also reached the same journal the authors now intend to publish their work. The authors show that it is possible to image missing O atoms due to the lattice mismatch in the intergrowth, and hence induces the formation of AIFR Lewis acid sites, which in the opinion of the authors are responsible for the observed promotion of the olefinic cycle; and hence results in more propylene and butene formation. While the STEM data are spatially resolved, the other methods used are bulk methods, focusing on the state of Al and acidity, as evaluated with NMR and IR with probe molecules. The novelty of the article is in my opinion in the use of the iDPC-STEM image to investigate in great detail an intergrowth structure of an industrially relevant zeolite, such as ZSM-5, while the other aspects are in my opinion important but less developed, worked.

Reply:

Thank the referee for the affirmation of our novelty in the use of the iDPC-STEM imaging to investigate an intergrown ZSM-5 architecture in great detail. Actually, our previous works (*Nature* **592**, 541 (2021); *Nat Commun* **12**, 2212 (2021); *Nat Commun* **11**, 2692 (2020); *Adv Mater* **32**, 1906103 (2020)) suggested that the technology of iDPC-STEM is an emerging, efficient tool for the atomic characterizations of local structures of beam-sensitive materials, even the confined single molecules. These results inspired us to resolve the intergrowth structures of ZSM-5-MT in this work.

2. As such the work deserves publication in a high-impact journal, but in its current version I cannot recommend the work for publication. First of all, I wish to stress that intergrowth structures can be present in different forms and shapes for zeolite ZSM-5 as shown in the 2009 Nature Materials of Karwacki, Kox and co-workers. While the works of a.o. Stavitski et al., Kox et al., and Roeffaers et al. report mainly the classical coffin-shaped intergrowth structure, interestingly there are reports out in which the "mortise-tenon" intergrowth structure has been shown to exist; an example can be found in the paper of Kox et al., *Chem. Eur. J.* 2008, 14, 1718-1725. It would be evaluate the proposed structure with the one described in this article; furthermore, one can also find examples, although not exactly the same in the paper of Mores et al., *Chem. Eur. J.* 2011, 17, 2874-2884.

Reply:

Thank you for your constructive comments. The "mortise-tenon" intergrowth structures with different forms are ubiquitous among ZSM-5 crystals, indicating that studying the structure-properties relationship of such intergrowth structures is important for developing efficient ZSM-5 catalysts. To this date, the atomic-scale information on such intergrowth structures lacks, owing to the limits of the low-dose imaging with

(scanning) transmission electron microscope ((S)TEM) by their sensitivity to electron beams, the low contrasts of light elements, and the lack of atomic-ordered nanocrystals. In this work, we for the first time atomically resolved the 90° intergrown structure in ZSM-5, and revealed its structure-properties relationship. Apart from the iDPC-STEM technique, the well-designed mortise-tenon ZSM-5 nanocrystal is also critical. Both the simple “mortise-tenon” intergrowth structure and nanosized ZSM-5 crystals with a short b-axis (50-70 nm) are necessary for atomically resolving.

The micro-sized intergrown ZSM-5 crystals (~ 10 μm) described in the papers of “Kox et al., Chem. Eur. J. 2008, 14, 1718-1725”, “Mores et al., Chem. Eur. J. 2011, 17, 2874-2884”, and “2009 Nature Materials of Karwacki, Kox and co-workers” are composed of many components (> 4), which make it hard to atomically resolving the intergrowth structures by electron microscope, much less revealing their structure-properties relationship. Fortunately, the nanosized mortise-tenon ZSM-5 crystals only consist of two components (a mortise, a tenon) and one interface.

We have added two references of “Kox et al., Chem. Eur. J. 2008, 14, 1718-1725” and “Mores et al., Chem. Eur. J. 2011, 17, 2874-2884” in the revised manuscript (*Ref. 39,40*).

3. What is important to realise is that the "woodworkers" concept is not that straightforward as proposed by the authors as the total penetration should not always be the case, nor can it be guaranteed that there is no Al zoning; and as I also expect that there is at the interface disorders/defects/missing atoms, hence Lewis acid sites, the bulk methods taken to evaluate this in the work are not fully conclusive. For this, the authors should have been used spatially resolved methods, although their crystals studied are smaller than those described in the above-mentioned references.

Reply:

Thank you very much for the constructive suggestions. Partial penetration is common in our ZSM-5 samples. In fact, both total and partial penetration can be considered as analogs of traditional mortise-tenon junctions.

(a) EDS analysis:

To confirm that there is no Al zoning, we performed energy-dispersive spectroscopy (EDS) analysis to determine the distribution of Al element in mortise-tenon ZSM-5 nanocrystals.

Fig. R1. The EDS mapping of ZSM-5-MT crystals from [001] direction (up) and the

lateral view (bottom).

The mappings of Al element in ZSM-5-MT crystals indicate that there is no obvious Al zoning in the whole intergrown crystal.

(b) Laser confocal fluorescence microscopy:

We have tried to characterize the intergrowth ZSM-5-MT using laser confocal fluorescence microscopy. The method we used was demonstrated in some pioneer works (*ACS Catal.* **7**, 4248–4252 (2017), *Nat. Comm.* **10**, 4348 (2019).). The furfuryl alcohol was used as the probe molecule, emitting strong fluorescence when catalyzed by Brønsted acid sites. When water was used as the solvent, the oligomer fluorophores preferentially occupied the straight channels. When the straight channels run parallel to the direction of polarized excitation light, the oligomer fluorophores in straight channels will show the strongest fluorescence emission. The areas without illumination are because the sinusoidal channels lack fluorophores or straight channel runs perpendicular to the polarized excitation light. In principle, the layer by layer depth scan profiles along the [010] direction of mortise subunit of ZSM-5-MT crystals can reveal the macro-shape of intergrowth in the interior of the mortise subunit.

Our results are presented in Fig. R2a and b, which shows that some blurred green dots appeared in images. In conclusion, laser confocal fluorescence microscopy is not available for these nanosized ZSM-5-MT crystals, because of its limited resolution. On the one hand, from [010] direction of the mortise subunit, the intergrowth area (e.g. 30 nm × 194.5 nm in Fig. R2c) is too small to be resolved by laser confocal fluorescence microscopy. On the other hand, the maximum depth of intergrowth in the interior of the mortise subunit is only ten nanometers (Fig. R2d), which is much smaller than the step size (several hundred nanometers) of depth scans along the z-axis.

These results further highlight that the iDPC-STEM technique is greatly powerful and advantageous for resolving the architectures of nanosized materials.

Fig. R2. (a, b) Confocal fluorescence microscopy of ZSM-5-MT crystals. (c, d) The ADF-STEM images of ZSM-5-MT crystals from different views.

(c) ADF-STEM images:

In the revised Supplementary Information (SI), we provided more ADF-STEM images of ZSM-5-MT from the lateral view. In Fig. R3, a gray curve inside the underlying crystal can be found in three individual ZSM-5-MT crystals. The atomic iDPC-STEM images in Fig. 2 have indicated the obvious missing O atoms at the intergrowth interfaces. And, Milward et al (*J. Chem. Soc., Faraday Trans.* 1983, 79, 1075) and D. G. Hay et al (*Zeolites* **10**, 571-576 (1990), *Ref.* 22) have proposed a model of such interface between 90°-intergrowth components, where the number of oxygen links across the boundary is half that in the ideal model. Thus, we concluded that these gray curves represent the boundary between the tenon crystal and underlying mortise crystal, which result from the missing O atoms at the intergrown interfaces. These observations further confirmed the intergrowth structures we proposed and provided another spatially resolved evidence for missing O atoms at the interface.

Fig. R3. The ADF-STEM images of three individual ZSM-5-MT crystals from the lateral view. The boundary between tenon and underlying mortise crystal marked by red arrows in magnified image. (Fig. R2 b and c have been added in Supplementary Figure 2 in revised SI)

4. Nevertheless, how sure are we that the differences are related to changes in LAS and not in changes in pore accessibility, number of BAS, as well as their strength. These points have to be addressed in a revised article before the paper can be published.

Reply:

Thank you for your constructive comments.

(a) The pore accessibility

For ZSM-5, the size of the straight channel ($5.3 \times 5.6 \text{ \AA}$) is very close to that of the sinusoidal channel ($5.1 \times 5.5 \text{ \AA}$). Again, Milward et al (*J. Chem. Soc., Faraday Trans.* 1983, 79, 1075) and D. G. Hay et al (*Zeolites* **10**, 571-576 (1990), *Ref. 22*) have proposed that the straight channels and the sinusoidal channels can be connected without any pore blocking. Besides, a tenon crystal vertically grows on only one of the (010) surfaces of the mortise crystal. Thus, the straight channels covered by the tenon crystal can be accessible from another (010) surface.

(b) Number of BAS and strength of BAS

As discussed in the content of “**Catalytic performances of ZSM-5-MT catalyst**” of our current manuscript, we compared the catalytic performances of ZSM-5-MT with ZSM-5-Sb-75 to demonstrate that the differences in catalytic performance are related to changes in LAS, since these two catalysts possess identical BAS numbers, which

was characterized by Py-FTIR (Table R1). To exclude the concern on acid strength, we performed NH₃-TPD experiments to characterize ZSM-5-MT and ZSM-5-Sb-75 catalysts. As shown in Fig. R3, the acid sites strength of these two catalysts are identical. We added NH₃-TPD profiles of two samples in SI. Thus, we carefully confirmed that the differences in catalytic performance are related to changes in LAS, and we think your concerns can be fully addressed.

In the revised content “**Catalytic performances of ZSM-5-MT catalyst**” of the manuscript, we have added “To address the effect of these Al_{FR} Lewis acid sites, we tested the catalytic performances of ZSM-5-MT and ZSM-5-Sb-75 catalysts with the same concentration **and strength** of acid sites in the conversion of methanol.”

Table R1. The acid site concentrations were determined from FT-IR spectra of adsorbed pyridine. (Supplementary Table 2 in SI)

Sample	Acid site concentration (μmol/g)	
	LAS	BAS
ZSM-5-MT	14.22	167.63
ZSM-5-Sb-75	14.80	143.30

Fig. R3. NH₃ temperature-programmed desorption (TPD) profiles for ZSM-5-MT (a) and ZSM-5-Sb-75 (b). (Fig. R3 has been added in Supplementary Figure 17 in revised SI)

REVIEWERS' COMMENTS

Reviewer #4 (Remarks to the Author):

As referee 4 I have now assessed the revised version of the paper and believe that the authors have attempted to address my main concerns on the initial version of their work. Although I do not yet believe that all the comments are sufficiently addressed (in terms of the Lewis acid/defect sites. etc.) I still believe the work is worth publishing and hence I recommend it for publication in Nature Communications.